Proceedings of the 6th Symposium on Advances in Approximate Bayesian Inference, 2024 1–32

# Implicitly Bayesian Prediction Rules in Deep Learning

**Bruno Mlodozeniec**[1,2]                                                            BKM28@CAM.AC.UK
**David Krueger**[1]                                                                  DSK30@CAM.AC.UK
**Richard Turner**[1]                                                                 RET26@CAM.AC.UK
[1]*University of Cambridge*        [2]*Max Planck Institute for Intelligent Systems*

## Abstract

The Bayesian approach leads to coherent updates of predictions under new data, which makes adhering to Bayesian principles appealing in decision-making contexts. Traditionally, integrating Bayesian principles into models like deep neural networks involves setting priors on parameters and approximating posteriors. This is done despite the fact that, typically, priors on parameters reflect any prior beliefs only insofar as they dictate function space behaviour. In this paper, we rethink this approach and consider what properties *characterise* a prediction rule as being Bayesian. Algorithms meeting such criteria can be deemed *implicitly* Bayesian — they make the same predictions as *some* Bayesian model, without explicitly manifesting priors and posteriors. We argue this might be a more fruitful approach towards integrating Bayesian principles into deep learning. In this paper, we propose how to measure how close a general prediction rule is to being implicitly Bayesian, and empirically evaluate multiple prediction strategies using our approach. We also show theoretically that agents relying on non-implicitly Bayesian prediction rules can be easily exploited in adversarial betting settings.

## 1. Introduction

In the Bayesian framework, model predictions are updated coherently and rationally based on new evidence. These rationality properties are epitomised by various theorems showing that a Bayesian agent states and updates their beliefs in a way that cannot be trivially exploited by an adversary (Pettigrew, 2020; Lane and Sudderth, 1984, §4). This arguably gives Bayesian predictions some level of credibility; e.g. when following the Bayesian approach exactly, one does not have to worry (as much) about whether some evidence is being given more weight than other, or whether some evidence is ignored outright. Even putting computational and approximate inference considerations aside, a Bayesian still has plenty to worry about with regards to modelling choices; nonetheless, at the very least, some fundamental sanity checks are taken care of.

This is in contrast to many other — potentially more "black-box" — approaches to prediction, such as training a deep learning model. For instance, when continually updating a deep learning model with new data, we might have to worry about catastrophic forgetting (McCloskey and Cohen, 1989), or that the order the data is presented in might adversely affect the predictions (Ash and Adams, 2020).

When a model makes predictions that will be acted upon in a closed decision-making loop, e.g. when finding the optimum of an unknown function (Bayesian Optimisation) or for exploration in reinforcement learning, it is important that the model's predictions change in a coherent way in light of new observed data. In these settings, the uncertainty estimates provided by the model are used to guide the decision-making process, often in a way that

has to balance exploration against exploitation. However, if the model's current predictions and uncertainty estimates are not related to how the model's predictions will change upon observing a currently unobserved variable, then any down-stream decisions made on the basis of these predictions might be highly suboptimal.

This, among other reasons, is often the reason Bayesian methods are being advocated for, and why many attempts have been made to incorporate Bayesian principles into deep learning. However, despite considerable effort by a large community, existing approaches to putting deep learning within a Bayesian framework often perform poorly in practical settings. Current research typically focuses on one specific model class, which is obtained by placing a prior over the parameters of the neural network, and approximating the posterior.[1]Algorithms relying on this perspective usually encumber scalability. For example, in Markov Chain Monte Carlo (MCMC) schemes, effectively, multiple models need to be trained following a cumbersome procedure for an accurate approximation; variational inference schemes, on the other hand, struggle to fit complex posteriors over the parameters without similarly complex and difficult to train models to approximate that posterior.

In this paper, we advocate for taking a different perspective on how to incorporate Bayesian principles into deep learning, and how to even think about measuring how close we are to achieving this goal. Specifically, we look at what desirable properties predictions made by Bayesian methods posses (including a new Dutch-book-style theorem in Section 3), what properties *characterise* a prediction rule as being Bayesian (Section 2.3), and how to measure how close a prediction rule is to satisfying these properties (Section 4). We empirically demonstrate our proposed measure by investigating how various design decisions affect how close an algorithm is to being implicitly Bayesian on a small regression task (Section 4.2). We advocate that this might be a more fruitful way to think about incorporating Bayesian principles into deep learning, as it only dictates a minimal set of conditions for how the predictions should behave, rather than dictating how the internals of the prediction algorithm should be structured.

## 2. Background

Whereas statistics often deals more broadly with inferences about various unobserved quantities, the prediction of future observations is arguably at the core of machine learning. Hence, in this piece, we primarily consider the setting of predicting future observations given the past. Concretely, given a sequence of random variables $X_1, X_2, \ldots$ we are interested in predicting the values of $X_{n+1}, X_{n+2}, \ldots$ given observations of $X_1, \ldots, X_n$ for different $n$. We'll also look at the case of regression/classification where for a sequence of random variables $X_1, X_2, \ldots$ and $Y_1, Y_2, \ldots$ we are interested in predicting the value of $Y_{n+1}$ given the observations of $X_1, Y_1, \ldots, X_n, Y_n$, and $X_{n+1}$.

To discuss and compare the properties of various approaches to predicting future observations, it is helpful to introduce the concept of a *prediction rule/strategy*. If the observations take values in some space $\mathcal{X}$, then (informally) a prediction rule is a sequence of functions $(s_0, s_1, s_2, \ldots)$ where each $s_n$ maps a sequence $(x_1, \ldots, x_n) \in \mathcal{X}^n$ to a probability distribution $s_n(\cdot | x_1, \ldots, x_n)$ on $\mathcal{X}$. $s_n(\cdot | x_1, \ldots, x_n)$ carries the interpretation of the

---

1. Function-space variational inference (Sun et al., 2019) being a notable exception, although in this case a prior is still specified explicitly, just directly in the function space.

prediction for the next observation $X_{n+1}$ given the observed outcomes $(x_1, \ldots, x_n)$ for the previous observations $X_1, \ldots, X_n$.

More precisely, Dubins et al. (2014) formally introduce a prediction rule/strategy on a measurable space $(\mathcal{X}, \mathcal{F})$ as a sequence of functions $s_n : \mathcal{F} \times \mathcal{X}^n \to [0, 1]$ where:

1. For every $(x_1, \ldots, x_n) \in \mathcal{X}^n$, the function $A \mapsto s_n(A|x_1, \ldots, x_n)$ for $A \in \mathcal{F}$ is a probability measure on $(\mathcal{X}, \mathcal{F})$,
2. For every $A \in \mathcal{F}$, the function $x_1, \ldots, x_{n-1} \mapsto s_n(A|x_1, \ldots, x_{n-1})$ is $\otimes_{i=1}^n \mathcal{F}$-measurable (with $\otimes_{i=1}^n \mathcal{F}$ denoting the product $\sigma$-algebra on $\mathcal{X}^n$)

These two conditions equivalently specify that each $s_k$ is a Markov kernel from $(\mathcal{X}^k, \otimes_{i=1}^k \mathcal{F})$ to $(\mathcal{X}, \mathcal{F})$, effectively ensuring that the prediction rules define a joint probability measure on $(\mathcal{X}^n, \otimes_{i=1}^n \mathcal{F})$, i.e. on the sequence of first $n$ observations, for all $n$. Furthermore, by the *Ionescu-Tulcea Theorem* (Hoffman-Jorgensen, 2017; Berti et al., 2023), a prediction rule uniquely defines a probability measure over the whole infinite sequence space $(\mathcal{X}^\infty, \otimes_{i=1}^\infty \mathcal{F})$, hence formally justifying using a prediction rule to make predictions on the whole sequence of observations $(X_1, X_2, \ldots)$; it also allows us to define properties of prediction rules in terms of the joint they imply over the sequence space.

A prediction rule describes how a practitioner makes predictions about the future observations given the past. In the context of deep learning, a prediction rule might encompass the whole procedure for training a neural network on a dataset of past observations, and then using the trained neural network to make predictions about future observations. For example, $s_n(x_{n+1}|x_1, \ldots, x_n)$ might be defined as the probability density of $x_{n+1}$ given by a normalising flow trained on a dataset of examples $(x_1, \ldots, x_n)$ following, for example, Stochastic Gradient Descent (SGD) with a maximum likelihood objective[2].

A standard assumption in machine learning problems is that the data $(X_1, X_2, \ldots)$ is independent and identically distributed (*i.i.d.*) — i.e. the random variables $(X_1, X_2, \ldots)$ are independent, and they all follow the same law: $X_i \sim P_X$. The *data generating distribution* $P_X$ is unknown to the practitioner. In this paper, we'll primarily concern ourselves with the case of *i.i.d.* data. We'll describe below how one would go about defining a prediction rule in the Bayesian framework in an *i.i.d.* context, and then look at various properties that such prediction rules might have.

## 2.1. The Bayesian Inferential Approach

In its most general form, the Bayesian framework for inference is to **1)** specify a joint distribution over all the random variables of interest, and **2)** condition on the observed values to obtain a posterior distribution over the unobserved variables of interest. In the context of a sequential prediction problem, one might specify a joint distribution over the random variables $(X_1, X_2, \ldots)$ and condition on the observed values of $X_1, \ldots, X_n$ to obtain a posterior distribution over e.g. $X_{n+1}$. For *i.i.d.* data, a Bayesian would usually treat the data generating distribution $P_X$ as an unknown, and place a prior distribution over it. To make predictions about the next observation $X_{n+1}$ given observed outcomes $(x_1, \ldots, x_n)$ for

---

2. In this framework, prediction rules are deterministic; to view training a deep learning model as a prediction rule, all sources of randomness other than the data have to be fixed (e.g. through the seed). Each seed or sequence of seeds effectively leads to a different prediction rule.

the previous observations $X_1, \ldots, X_n$, one could describe the Bayesian *inferential* approach (Berti et al., 2023) as: **I.** Specify a prior over the generating distribution; **II.** Get a posterior over the generating distribution given the observed data; **III.** Compute the prediction $s_n(\cdot|x_1, \ldots, x_n)$ by computing the posterior predictive distribution for $X_{n+1}$.

For instance, assuming the distributions of interest can be described with a parameter $\theta \in \mathbb{R}^d$ and a density $p_{X|\Theta}(x|\theta)$, and that the prior over $\Theta$ also has a density $p_\Theta(\theta)$, the procedure above might look like:

I Specify a prior density $p_\Theta(\theta)$ over the parameters;

II Compute the posterior density $p_{\theta|X_1,\ldots,X_n}(\theta|x_1, \ldots, x_n) \stackrel{\text{def}}{\propto} \prod_{i=1}^n p_{X|\theta}(x_i|\theta)p_\theta(\theta)$;

III Calculate the prediction $s_n(\cdot|x_1, \ldots, x_n) \stackrel{\text{def}}{=} \int p_{X|\theta}(\cdot|\theta)p_{\theta|X_1,\ldots,X_n}(\theta|x_1, \ldots, x_n)d\theta$

### 2.2. Implicitly Bayesian Prediction Rules

An alternative to the inferential approach would to be to specify a prediction rule directly. This approach is often referred to in the literature as the *predictive approach* (Berti et al., 2023) and has been recently studied extensively in the statistics literature (Berti et al., 2013; Fong, 2021; Fong and Lehmann, 2022; Berti et al., 2019, 2021, 1998). It should be evident that this procedure *can* in effect result in the same prediction rule as that from the inferential approach.

For example, the practitioner could specify a linear model on $(X_n, Y_n)$ with a uniform prior density on the covariates $X_n$ (say, in the range $[0, 1]^d$), and a Gaussian prior $\mathcal{N}(\theta; 0, I)$ on the weights $\theta$ (assuming homogeneous Gaussian noise with variance $\sigma^2$). Here $y \mapsto \mathcal{N}(y; \mu, \Sigma)$ denotes a Gaussian density with mean $\mu$ and covariance $\Sigma$. Given observations $((x_1, y_1), \ldots, (x_n, y_n))$, the practitioner would then construct a posterior on the weights (e.g. through Monte-Carlo sampling) and average over the posterior samples to obtain a posterior predictive distribution for $X_{n+1}, Y_{n+1}$. Alternatively, they could directly compute an equivalent prediction, without directly manifesting the posterior, with the prediction rule given in Appendix C. If a practitioner happened, by a stroke of luck, to specify this as their prediction rule without ever considering the underlying assumptions of a linear model and a prior, they'd still make the same predictions as if they followed the Bayesian framework with some underlying model.

A natural question to ask is: under what conditions on the prediction rule is it equivalent to the inferential approach for *some* prior and likelihood? In other words, given a prediction rule, can one say whether there exists a prior and likelihood such that the predictions from the prediction rule match those of following the Bayesian framework with that likelihood/prior pair under the *i.i.d.* assumption? If this is the case, we'll say that the prediction rule is *implicitly Bayesian*.[3]

In what follows, we'll look at the properties that the prediction rules defined following the Bayesian framework posses, and the properties that characterise them.

### 2.3. Characterising Implicitly Bayesian Prediction Rules

In the case of assumed *i.i.d.* data, De Finetti's theorem gives a simple condition for a prediction rule to be implicitly Bayesian. As mentioned before, a prediction rule implies a

---

3. The prediction rule and the corresponding likelihood/prior construction can be said to be *Bayes-ically the same*.

unique joint distribution over the sequence of random variables $(X_1, X_2, \dots)$. By a version of the De Finetti's theorem, under some mild assumptions, a prediction rule is implicitly Bayesian if and only if the joint distribution it implies over $(X_1, X_2, \dots)$ is *exchangeable* (Hewitt and Savage, 1955):

**Definition 1 (Exchangeable Sequence of Random Variables)** *A finite sequence of $n$ random variables $(X_1, \dots, X_n)$ is said to be exchangeable if for any permutation $\pi :$ $\{1, \dots, n\} \to \{1, \dots, n\}$ the joint distribution of $(X_1, \dots, X_n)$ is the same as the joint distribution of $(X_{\pi(1)}, \dots, X_{\pi(n)})$.*

*An infinite sequence of random variables $(X_1, X_2, \dots)$ is said to be exchangeable if, for any $n$, the finite sequence $(X_1, \dots, X_n)$ is exchangeable.*

By De Finetti's theorem (see Appendix A for a more formal introduction), we know that a sequence of random variables $(X_1, X_2, \dots)$ is exchangeable if and only if there exists a (unique) prior probability $\pi$ on the space $\mathcal{P}$ of probability measures on $\mathcal{X}$ such that:

$$P[X_1 \in A_1, \dots, X_n \in A_n] = \int_{\mathcal{P}} \prod_{i=1}^{n} p(A_i) d\pi(p) \qquad \forall A_1, \dots, A_n \forall n \in \mathbb{N},$$

where $p \in \mathcal{P}$ is a probability measure on $\mathcal{X}$. In other words, only if the there exists a likelihood/prior construction that defines the same joint distribution as the prediction rule. Hence, exchangeability is the defining characteristic of implicitly Bayesian prediction rules on *i.i.d.* data. This suggests one direct way of checking whether a prediction rule is implicitly Bayesian: check whether the joint distribution on $(X_1, X_2, \dots)$ implied by the prediction rule is exchangeable.

**Conditionally Identically Distributed** Another desirable coherence property that we might expect of a prediction rule is that the future observations are identically distributed given the past. For example, under the *i.i.d.* assumption, if we were to observe $(x_1, \dots, x_n)$, the prediction for the next observation $X_{n+1}$ *surely* shouldn't be different from the prediction for the observation after that. After all, we know they are identically distributed, we just don't know what the distribution is; it'd be irrational to make different predictions for $X_{n+1}, X_{n+2}$ given the same data.

This property can be formalised as follows:

**Definition 2 (Conditionally Identically Distributed)** *We say that a sequence of random variables $(X_1, X_2, \dots)$ is conditionally identically distributed (c.i.d.) if for any $n$:*

$$P[X_{n+1} \in \cdot | X_1 = x_1, \dots, X_n = x_n] = P[X_{n+k} \in \cdot | X_1 = x_1, \dots, X_n = x_n] \quad \forall k > n$$

*holds almost surely.*

A prediction rule is then *c.i.d.* if the joint distribution it implies over $(X_1, X_2, \dots)$ is *c.i.d.*. Conditionally Identically Distributed sequences have been introduced in (Kallenberg, 1988) and studied and applied in a range of works (Berti et al., 2004, 2013; Fong et al., 2021a).

It should be clear that each exchangeable sequence is *c.i.d.*, as exchangeability implies $X_1, \dots, X_n, X_{n+1} \stackrel{d}{=} X_1, \dots, X_n, X_{n+k}$ for any $k > 0$. Hence, following the Bayesian framework will yield *c.i.d.* prediction rules, and any implicitly Bayesian prediction rule will be *c.i.d.*. Not all *c.i.d.* sequences are exchangeable, however. *c.i.d.* can hence be seen as a weakening of the condition of exchangeability.

**Stationarity** There is one complimentary property that not all *c.i.d.* sequences have that would make them *implicitly Bayesian*. Namely, *stationarity*:

**Definition 3 (Stationary sequence)** *A sequence of random variables* $(X_1, X_2, \dots)$ *is said to be stationary if for any* $n, k > 0$:

$$X_1, \dots, X_n \overset{d}{=} X_{1+k}, \dots, X_{n+k} \tag{1}$$

In the context from the prediction rules, stationarity implies sampling $k$ datapoints $x'_1, \dots, x'_k$, from the prediction rule itself and prepending them to the observed data $\mathcal{D} = (x_1, \dots, x_n)$, would yield a prediction rule $s_{n+k}(\cdot | x'_1, \dots, x'_k, \mathcal{D})$ that's in expectation the same as $s_n(\cdot | x_1, \dots, x_n)$.

By the result of Kallenberg (1988, Proposition 2.1), exchangeability exactly amounts to stationarity and the *c.i.d.* condition; stationarity and *c.i.d.* properties are another way of characterising implicitly Bayesian prediction rules.

**Spreadability** Another condition that turns out to also characterise implicitely Bayesian models is *spreadability*:

**Definition 4 (Spreadable sequence)** *A sequence of random variables* $(X_1, X_2, \dots)$ *is said to be spreadable if for any* $n > 0$ *and any sequence of indices* $k_n > k_{n-1} > \cdots > k_1 \geq 1$:

$$X_1, \dots, X_n \overset{d}{=} X_{k_1}, \dots, X_{k_n} \tag{2}$$

Spreadability says that the distribution of the first $n$ observations is the same as the distribution of any $n$ observations from the sequence. In the context of a prediction rule, it implies that, if we were to construct a new dataset by possibly sampling $k_i \geq 0$ datapoints from the prediction rule conditioned on the previous datapoints inbetween "observed" datapoints $x_i, x_{i+1}$, the final prediction given that dataset would in expectation be the same as the prediction given the original dataset. Kallenberg (1988) has shown that spreadability is equivalent to exchangeability. In other words,

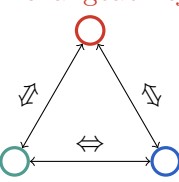

Figure 1: Equivalent characterisation of exchangeable sequences, and hence implicitly Bayesian prediction rules.

spreadability is yet another property that characterises implicitly Bayesian prediction rules.

In summary, Figure 1 illustrates the different properties that an implicitly Bayesian prediction rule posses and can be characterised by. Some might be easier to establish or approximately enforce then others, and each suggests its own way of constructing an exchangeable prediction rule from a non-exchangeable one. It's an interesting question as to what extent various commonly used machine learning and deep learning methods posses these different properties, and how various design choices might affect them.

**Can a prediction rule at a fixed step be extended to an implicitly Bayesian prediction rule?** In deep learning, we would often taylor the approach to the amount of data available at hand. If we have fewer than hundreds of datapoints, we might not even consider using a neural network at all. One might therefore wonder, given a specification for a prediction rule at step $n$ only, can it be extended to a prediction rule for all $n$ to satisfy the aforementioned properties? We discuss this in Appendix E, but, in short, the requirements on $s_n$ turn out to be quite stringent.

## 3. Non-implicitly Bayesian agents are vulnerable to adversarial bets

Many of the fundamental axioms of probability, and hence probabilistic inference, can be argued for on the basis of various Dutch Book arguments (Pettigrew, 2020). These show that, unless one adopts the intuitively sensible-seeming axioms of probability (e.g. probabilities must sum to 1), one is left vulnerable to accepting seemingly appealing, but highly disadvantageous, bets. In this subsection, along similar lines, we'll show that using a non-exchangeable prediction rule, when the true distribution is exchangeable (e.g. *i.i.d.*), leaves one vulnerable to accepting bets that are in expectation disadvantageous. We give a brief overview of the setup and the result here, but for a thorough exposition and the proof see Appendix B.

Consider a game in which two players (an agent and an adversary) make bets on a sequence of $n$ observations from some unknown exchangeable or *i.i.d.* distribution. Here, we'll restrict ourselves to the finite discrete setting where the sequence of events takes values in $\mathcal{X}^n$ for some finite space $\mathcal{X}$. The agent and the adversary agree ahead of time how much the adversary will pay to (or receive from) the agent for every possible observed outcome. This is captured in a bet function $r : \mathcal{X}^n \to \mathbb{R}$, which represents the reward of $r(x_{1:n})$ to the agent, and cost of $-r(x_{1:n})$ to the adversary, when the observed outcomes is $x_{1:n} \in \mathcal{X}^n$.

The agent and adversary are taken to have some 'beliefs' over what the observations might be. The beliefs are taken to be probability mass functions on $\mathcal{X}^n$. Their beliefs dictate what kind of bets they are willing to accept. Given agent and adversary beliefs $q : \mathcal{X}^n \to \mathbb{R}$ and $\bar{q} : \mathcal{X}^n \to \mathbb{R}$, a bet $r : \mathcal{X}^n \to \mathbb{R}$ is said to be *admissible* if it appears 'favourable' from the point of view of both the agent and the adversary, namely if $\sum_{x_{1:n} \in \mathcal{X}^n} r(x_{1:n}) \bar{q}(x_{1:n}) > 0$ and $\sum_{x_{1:n} \in \mathcal{X}^n} r(x_{1:n}) \bar{q}(x_{1:n}) < 0$. These inequalities can be interpreted as saying, from the point of view of the beliefs of the agent and the adversary, their respective expected payoffs are positive. It is also helpful to define a notion of a *minimal* bet: given agent and adversary beliefs, a bet $r$ is minimal if it is the smallest possible[4] among all bets that give the same expected return to both the agent and the adversary (see Appendix B for details). Restricting to minimal bets ensures that the players do not arbitrarily make bets that are not justified on the ground of their beliefs.

We will consider an adversary that constructs an "exchangeable-ified" version $\bar{q}$ of the agent's beliefs $q$ as:

$$\bar{q}(x_1, \ldots, x_n) = \frac{1}{n!} \sum_{\pi \in \Pi_n} q(x_{\pi(1)}, \ldots, x_{\pi(n)}), \tag{3}$$

where $\Pi_n$ is the collection of all permutations of $n$ elements. It should be clear that $\bar{q}$ is itself an exchangeable probability mass function.

We provide a specific construction demonstrating that any agent following non-exchangeable beliefs (e.g. those implied by a non-exchangeable prediction rule) is necessarily exploitable, at least by an agent with an "exchangeable-ified" version of their beliefs. This is formalised in the following theorem:

**Theorem 5** *Given an agent with beliefs $q : \mathcal{X}^n \to [0,1]$, and an adversary with beliefs $\bar{q}$ that are an "exchangeable-ified" version of beliefs of the agent $\bar{q}$ as defined in (3), for any*

---

4. Smallest in $\ell_2$ norm, where the $\ell_2$ norm of $f : \mathcal{X}^n \to \mathbb{R}$ is taken to be $\sqrt{\sum_{x_{1:n} \in \mathcal{X}^n} f(x)^2}$.

*exchangeable distribution* $p : \mathcal{X}^n \to [0, 1]$ *that has some common support with* $q$, *all minimal and admissible bets* $r$ *have a strictly negative expected return for the agent under* $p$:

$$\sum_{x_{1:n} \in \mathcal{X}^n} p(x_{1:n})r(x_{1:n}) < 0 \tag{4}$$

*Furthermore, either:* **1)** $q$ *is exchangeable and* $\bar{q} = q$, *and there are* no *admissible bets; or* **2)** $q$ *is not exchangeable and there exist admissible (and minimal) bets.*

In particular, the above holds when $p$ is an *i.i.d.* distribution.

## 4. Measuring Implicit Bayesianness

**Notation** As we'll deal with nested expectations and variances in this section, we'll adopt the machine learning notation for conditional expectations: for a function $f$ that depends on multiple random variables $A, B, \ldots$ we'll write $\mathbb{E}_A[f(A, B, \ldots)]$ to denote the *conditional expectation* of $f$ conditioned on all the variables other than $A$ – the subscript indicates the "marginalised out" variables.[5] The subscript in the variance $\mathrm{Var}_A[f(A, B, \ldots)]$ is defined analogously.

In the preceding section, we argued that implicitly Bayesian prediction rules might be desirable, and presented various testable properties that characterise them. In this section, we'll look at how we can go about measuring these properties in practice, present empirical results for both Bayesian and non-Bayesian models including deep learning models, and show that simple design choices can lead to more or less implicitly Bayesian prediction rules. As shown in Section 2.3, exchangeability is one defining characteristic of implicitly Bayesian prediction rules. In this paper, we'll focus deriving empirical measures of approximate exchangeability, although the other criteria in Section 2.3 are equally interesting candidates.

In a machine learning context, we're usually dealing with settings where the observations take values either in a Euclidean or a discrete space, and the distribution over the future observations is derived from a conditional probability density or mass function. In this section, we will restrict ourselves to those two settings, and define a prediction rule as a sequence of functions $(s_0, s_1, s_2, \ldots)$ where each $s_n$ maps a sequence $(x_1, \ldots, x_n) \in \mathcal{X}^n$ to a probability density/mass function $s_n(\cdot | x_1, \ldots, x_n)$, overloading the notation above. Of course, such prediction rules can be converted to the more general definition above.

Exchangeability is not easy to verify. Even for a finite sequence, it requires checking that $\prod_{i=0}^{n-1} s_i(x_{i+1} | x_1, \ldots, x_i) = \prod_{i=0}^{n-1} s_i(x_{\pi(i+1)} | x_{\pi(1)}, \ldots, x_{\pi(i)})$ with probability 1 (i.e. for almost every sequence $(x_1, \ldots, x_n) \in \mathcal{X}^n$ with respsect to the joint measure on $\mathcal{X}^n$ implied by the prediction rule) for every permutation $\pi$. Checking the exchangeability condition for even for just a single sequence $(x_1, \ldots, x_n)$ can be computationally expensive, as it requires verifying the above equality for all $n!$ permutations of the sequence. Lastly, we rarely expect exact exchangeability to hold in practice. Even for implementations of models with a strong Bayesian motivation, the exact equality might not hold due to numerical errors or approximate inference. Ideally, we'd like some measure of the *degree* of exchangeability of a prediction rule.

To measure how close to being exchangeable a prediction rule is, in this paper we suggest measuring the *variance* of the log-joint as we randomly sample permutations of the data

---

5. For example, if $A, B, C$ are random variables, then $\mathbb{E}_A[\mathbb{E}_B[f(A, B, C)]] = \mathbb{E}[\mathbb{E}[f(A, B, C) | A, C] | C]$.

uniformly at random. Concretely, if $\Pi$ is a random variable that takes values in the set of all permutations of $\{1, \ldots, n\}$ with equal probability, we propose measuring:

$$\mathrm{Var}_\Pi \left[ \log \prod_{i=0}^{n-1} s_i(x_{\Pi(i+1)}|x_{\Pi(1)}, \ldots, x_{\Pi(i)}) \right] = \mathrm{Var}_\Pi \left[ \sum_{i=0}^{n-1} \log s_i(x_{\Pi(i+1)}|x_{\Pi(1)}, \ldots, x_{\Pi(i)}) \right]$$

for a given sequence of datapoints $(x_1, \ldots, x_n)$. Taking the log makes the measure more numerically stable. To get around measuring the variance for every sequence $(x_1, \ldots, x_n)$, we propose sampling sequences of datapoints and reporting the expected variance of the log-joint as a measure of implicit Bayesianness. Since it's nontrivial to evaluate the variance of the log-joint on an infinite sequence, we resort to checking for exchangeability on a finite sequence of datapoints. Arguably, we might care about 'being implicitly Bayesian' in some regions of the $\mathcal{X}^n$ space more than others. Hence, we evaluate the variance of the log-joint preferentially on sequences of data sampled from a task of interest. We resort to measuring the *expected* variance given some reference distribution $\mathcal{D}^n$ over sequences of $n$ datapoints.For a distribution $\mathcal{D}^n$ with full support, the prediction rule $s$ is exchangeable (on the first $n$ observations) if and only if the variance of the log-joint is zero, justifying the use of the variance of the log-joint as a measure of exchangeability. Let $X_1, \ldots, X_n$ be random variables that follow the law $\mathcal{D}^n$. Then, our measure looks like:

$$m_{\texttt{var}}((s_1, s_2, \ldots)) = \mathbb{E}_{X_1, \ldots, X_n} \left[ \mathrm{Var}_\Pi \left[ \sum_{i=1}^{n} \log s_i(x_{\Pi(i)}|x_{\Pi(1)}, \ldots, x_{\Pi(i-1)}) \right] \right] \tag{5}$$

One last caveat remains for the case of non-deterministic prediction rules such as training of a deep learning model. In this case, each random seed $\epsilon$ will yield a different prediction rule $s^\epsilon = (s_1^\epsilon, s_2^\epsilon, \ldots)$. Hence, the metric that we actually measure is the average variance of the log-joint over different random seeds $\epsilon \in \mathcal{E}$: $\frac{1}{|\mathcal{E}|} \sum_{\epsilon \in \mathcal{E}} m_{\texttt{var}}((s_1^\epsilon, s_2^\epsilon, \ldots))$. In practice, we approximate (5) by Monte-Carlo sampling, taking the empirical variance and mean of the log-joint to arrive at a computable metric.

### 4.1. Implicit Bayesianness vs. Performance

Although we argue implicit Bayesianness is a desirable property, it is not an end-goal on its own. After all, if a prediction rule has lacklustre performance on the tasks we are interested in, it likely won't be much comfort that it is implicitly Bayesian.[6] Hence, since predictive performance is a key consideration, we'll report both the aforementioned measure of implicit Bayesianness in (5), as well as a measure of performance on the data-generating distribution $\mathcal{D}^n$ in the experiments that follow. To report an aggregate of the performance on the entire sequence of $n$ observations, we'll report the average negative log-likelihood (NLL) of the prediction rule on the sequence of observations: $\mathbb{E}_{X_1, \ldots, X_n} \left[ \sum_{i=1}^{n} \log s_i(x_i|x_1, \ldots, x_{i-1}) \right]$. If the true data-generating distribution $\mathcal{D}^n$ is *i.i.d.*, which it will be for all the tasks considered below, then that expectation is the same as:

$$\mathbb{E}_{X_1, \ldots, X_n} \left[ \mathbb{E}_\Pi \left[ \sum_{i=1}^{n} \log s_i(x_{\Pi(i)}|x_{\Pi(1)}, \ldots, x_{\Pi(i-1)}) \right] \right] \tag{6}$$

---

6. It's not difficult to construct trivial implicitly Bayesian prediction rules. For example, one could set $s_i(x_i|x_1, \ldots, x_{i-1}) = q(x_i)$ for some fixed probability density/mass function $q : \mathcal{X} \to \mathbb{R}$. This prediction rule is trivially exchangeable, but it's not particularly useful as no learning is taking place.

allowing as to use the same samples to compute **1)** a measure of implicit Bayesiannes, i.e. the variance of the log-joint in (5) and **2)** a measure of performance, i.e. the expectation of the log-joint in (6). We investigate these metrics empirically in the following section.

### 4.2. Results

We consider a simple 1D regression task on which we compare deep learning prediction rules against exact and approximate Bayesian methods, such as exact conditioning in a Gaussian Process or approximate inference in Bayesian Neural Networks (BNNs). The task is pictured in Figure 12. The true function was chosen to be discontinuous to yield a model mismatch for a Gaussian Process with a smooth kernel. Figure 2 summarises all results.

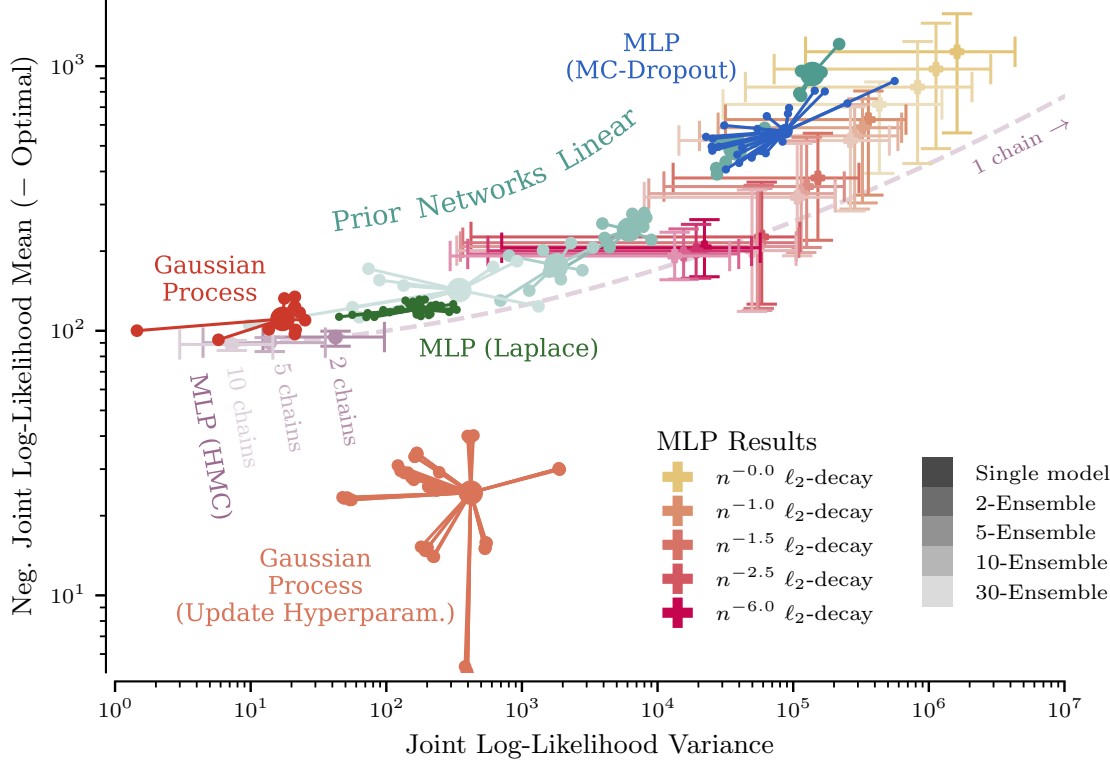

Figure 2: Evaluation of the joint log-likelihood variance as a measure of *implicit Bayesianness* (defined in (5)) vs. the negative sum of log-likelihood as a measure of *performance* (defined in (6)) on the regression task described in Section 4.2. The large dots represent the average over multiple dataset samples from the data-generating distribution, and the small dots/error-bars represent deviation of results for individual dataset samples $((x_1, y_1), \ldots, (x_{100}, y_{100}))$. For some methods, where ensembling multiple independently trained models might be of interest, we report results for multiple ensemble sizes; the lightness of the colour indicates the ensemble size. The expected negative log-likelihood of the *optimal* predictions on this dataset (using the true distribution) is subtracted from the mean negative joint log-likelihood plotted.

**Gaussian Process & Prior Network** To illustrate that a parametric model fit with gradient-descent on an objective function can be implicitly Bayesian, we compare a Gaussian Proccess (GP) against linear models fit with gradient descent following the "prior networks"

procedure described in Osband et al. (2018); this procedure entails full-batch gradient descent optimisation of the parameters on a negative log-likelihood objective (similarly to the canonical recipe for applying deep learning to regression), but with the targets augmented with random Gaussian noise and the weights $\ell_2$-regularised towards a random Gaussian sample. Osband et al. (2018) show this yields exact samples from the posterior of a Bayesian Linear Model. Hence, as we ensemble the predictions from more and more of these models, the resulting prediction rule should, in the limit, be implicitly Bayesian. To make the comparison clear, we chose the features and the (implicit) prior for the linear model in such a way that the resulting limiting model would be equivalent to the Gaussian Process with a squared exponential kernel its compared against (Appendix F.1).

The results for a Gaussian Process and prior networks are shown in Figures 4 and 2**a**. As expected, the linear model approaches both the performance and the implicit Bayesianness of the GP as the ensemble size increases. Notably, even the exact conditioning Gaussian process is not perfectly implicitly Bayesian due to numerical precision errors, as evidenced by the non-zero variance of the log-joint. Nonetheless, this result demonstrates that it's possible to get close to the implicit Bayesianness of exact Bayesian methods with algorithms that resemble those used in deep learning. It hopefully illustrates that improving implicit Bayesianness of deep learning algorithms might be an achievable task.

Motivated by the results above, we ask: what strategies or design choices might be most effective at improving implicit Bayesianness of deep learning models? Do methods with a Bayesian motivation necessarily outperform ones without?

**Ensembling of deep learning models** We run the same experiment as above with a deep learning model – a 3-hidden layer multi-layer perceptron (MLP) optimised with Stochastic Gradient Descent (SGD). We compare a base MLP model against ensemble predictions (Lakshminarayanan et al., 2017). The results are highlighted in Figures 5 and 6. Ensembling appears to not only improve predictive performance, but also improve the measure of implicit Bayesianness, although not by as much as for the linear model.

$\ell_2$ **Decay Schedules** A concern when defining a prediction rule by training a deep learning model for a fixed number of epochs is that the algorithm has effectively no notion of what the data-set size is. Training on two different datasets, the second one being a copy of the first one with each element repeated twice, would yield an identical training routine (bar effects of the random seed). A common workaround is to use stronger regularisation on smaller datasets. For example, the Maximum-a-Posteriori (MAP) estimation perspective of optimising the negative log-likelihood loss with $\ell_2$-regularisation suggests that the $\ell_2$ decay coefficient should be decayed as $n^{-1}$ (Bishop, 2006). Can a strategy as simple as decaying the $\ell_2$ regularisation coefficient as a function of the number of datapoints seen have a notable effect on the implicit Bayesianness of the resulting prediction rule? To investigate this, we run the same experiment as above, but with a decay schedule for the $\ell_2$ regularisation coefficient of the form $c_\alpha n^{-\alpha}$ for different values of $\alpha$. $c_\alpha$ is in each case set so that the value of $\ell_2$ decay would match for $n = 100$ (see Appendix F). The results are shown in Figure 7. Surprisingly, the improvements to both implicit Bayesianness and predictive performance are quite substantial, and, on this task, greater than what's achievable with ensembling of up to 10 models.

**Approximate Inference in BNNs** We also compare against various popular methods to approximate Bayesian inference in neural networks with a prior placed on the parame-

ters. We consider MC Dropout (Gal and Ghahramani, 2016), the Laplace approximation (Daxberger et al., 2022), and exact Hamiltonian Monte-Carlo (HMC) sampling. We also evaluated (non-linear) prior networks (Osband et al., 2018); the results are shown in Figure 8, but weren't included in the main Figure 2 as they were particularly noncompetitive. Interestingly, methods with a Bayesian motivation don't necessarily perform better than ones without. MC-Dropout is outperformed by regular SGD trained neural networks with a well-chosen $\ell_2$-decay schedule. The simple Laplace approximation performs surprisingly well, coming out ahead of even a Gaussian Process with kernel hyperparameters updated with empirical Bayes in terms of implicit Bayesianness.

## 5. Future Work

**Measuring other properties of implicitly Bayesian predictions** In this work, we only experimentally considered measuring exchangeability of a prediction rule. As discussed in Section 2.3, there are other desirable properties that an implicitly Bayesian prediction rules posses, and other conditions that characterise them. It would be interesting to investigate empirical metrics based on these different conditions.

**Martingale Posterior Sampling** As shown in (Fong et al., 2021b), if a prediction rule is *c.i.d.*, we can obtain a functional uncertainty estimates that give a notion of reducible uncertainty using *martingale posterior sampling*. Although the method in Fong et al. (2021b) would be computationally extremely burdensome for deep learning-based prediction rules, it would be interesting to investigate whether the method could be tractably adapted to this setting. Furthermore, it's an interesting question as to whether improving any of the implicit Bayesian properties would then lead to more *useful* uncertainty estimates, for example when used in Bayesian optimisation, active learning or reinforcement learning contexts.

**Searching for implicitly Bayesian updates** Turning implicit Bayesianness into a differentiable measure of closeness opens the doors for black-box optimisation for that property. Prior work has successfully meta-learned optimisers for faster training of more performant deep learning models (Metz et al., 2022). By meta-optimising update rules for *both* good performance and implicit Bayesianness, training algorithms that are both performant **and** update their predictions in a coherent way could perhaps be learnt.

## 6. Conclusion

In conclusion, this paper proposes a new perspective on incorporating Bayesian principles into deep learning, shifting focus from explicit model specification to characterising and achieving predictions that are implicitly Bayesian.

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

## Appendix A. De Finetti's theorem

Consider a probability space $(\Omega, \mathcal{F}_\Omega, P)$ and a sequence of random variables $(X_1, X_2, \dots)$, $X_i : \Omega \to \mathcal{X}$, where $X_i$ takes values in some measurable space $(\mathcal{X}, \mathcal{F})$. Then, by a version of the De Finetti's theorem given by Hewitt and Savage (1955), under fairly general conditions, if $(X_1, X_2, \dots)$ is exchangeable as per Definition 1, then the distribution of $(X_1, X_2, \dots)$ can be represented as a mixture of *i.i.d.* distributions.

Concretely, consider the set $\mathcal{P}$ of probability measures $p \in \mathcal{P}$ on $(\mathcal{X}, \mathcal{F})$, and the smallest $\sigma$-algebra $\mathcal{S}$ that makes $p \mapsto p(A)$ measurable for all $A \in \mathcal{F}$. For $p \in \mathcal{P}$, denote by $p^\infty$ the product probability measure $\otimes_{i=1}^\infty p$ on the product space $(\prod_{i=1}^\infty \mathcal{X}, \otimes_{i=1}^\infty \mathcal{F})$ (Saeki, 1996). Then, if $(X_1, X_2, \dots)$ is exchangeable, there exists a unique probability measure $\pi$ on $(\mathcal{P}, \mathcal{S})$ such that the joint distribution of $(X_1, X_2, \dots)$ can be represented as:

$$P[(X_1, X_2, \dots) \in A] = \int_{\mathcal{P}} p^\infty(A) d\pi(p) \qquad \forall A \in \otimes_{i=1}^\infty \mathcal{F}, \tag{7}$$

The only condition is that $\mathcal{X}$ must be a Hausdorff space with $\mathcal{F}$ being the $\sigma$-algebra of all Baire sets in $\mathcal{X}$. The practically relevant aspect of that condition is that it is satisfied in the typically considered settings of $\mathcal{X}$ being $\mathbb{R}$ or $\mathbb{R}^d$ with the usual Borel $\sigma$-algebra.

The above formulation is slightly different than the one given in Section 2.3. Namely, the proposition in (7) was instead replaced with: there exists a unique probability measure $\pi$ on $(\mathcal{P}, \mathcal{S})$ such that:

$$P[X_1 \in A_1, \dots, X_n \in A_n] = \int_{\mathcal{P}} \prod_{i=1}^n p(A_i) d\pi(p) \qquad \forall A_1, \dots, A_n \in \mathcal{F}, \forall n \in \mathbb{N}, \tag{8}$$

which we think is a little bit more approachable. The propositions in (7) and (8) are equivalent. Clearly, (7) implies (8); if the equality holds for all sets $A$ in the infinite product $\sigma$-algebra $\otimes_{i=1}^\infty \mathcal{F}$, then it will hold for rectangles $A = (A_1, \dots, A_n, \mathcal{X}, \mathcal{X}, \dots) \in \otimes_{i=1}^\infty \mathcal{F}$, and so:

$$
\begin{aligned}
P[X_1 \in A_1, \dots, X_n \in A_n] &= P[(X_1, X_2, \dots) \in A] \qquad B = (A_1, \dots, A_n, \mathcal{X}, \mathcal{X}, \dots) \\
&= \int_{\mathcal{P}} p^\infty(B) d\pi(p) \\
&= \int_{\mathcal{P}} \prod_{i=1}^n p(A_i) \prod_{i=n+1}^\infty \overbrace{p(\mathcal{X})}^{1} d\pi(p) \quad \triangle \text{Definition of product measure} \\
&= \int_{\mathcal{P}} \prod_{i=1}^n p(A_i) d\pi(p)
\end{aligned}
$$

To go the other way around, from (8) to (7), assume that there exists a unique measure $\pi$ such that (8) holds. We can extend the statement to 'infinite rectangle' sets of the form

$B = (A_1, A_2, \dots), A_i \in \mathcal{F} \; \forall i \in \mathbb{N}$ by noting that since $(A_1, \dots, A_n, \mathcal{X}, \mathcal{X}, \dots) \downarrow B$, we have that $P[X_1 \in A_1, \dots, X_n \in A_n] \downarrow P[(X_1, X_2, \dots) \in B]$. Hence:

$$
\begin{aligned}
P[(X_1, X_2, \dots) \in B] &= \lim_{n \to \infty} P[X_1 \in A_1, \dots, X_n \in A_n] \\
&= \lim_{n \to \infty} \int_{\mathcal{P}} \prod_{i=1}^{n} p(A_i) d\pi(p) \\
&= \int_{\mathcal{P}} \lim_{n \to \infty} \prod_{i=1}^{n} p(A_i) d\pi(p) \quad \triangle \text{Dominated Convergence Theorem} \\
&= \int_{\mathcal{P}} p^{\infty}(B) d\pi(p)
\end{aligned}
$$

Now, since $P[(X_1, X_2, \dots) \in \cdot]$ and $\int_{\mathcal{P}} p^{\infty}(\cdot) d\pi(p)$ are two measures on $(\prod_{i=1}^{\infty} \mathcal{X}, \otimes_{i=1}^{\infty} \mathcal{F})$ that agree on the $\pi$-system of rectangles that generates the product $\sigma$-algebra, they must agree on the entire $\sigma$-algebra.

It is worth noting that exchangeability of the entire infinite sequence $(X_1, X_2, \dots)$ is required for the result to hold. One might hope that if a finite sequence $(X_1, \dots, X_n)$ is exchangeable for some $n$, then the joint distribution of these $n$ random variables might also be a mixture distribution. This is not always the case, see e.g. (Kerns and Székely, 2006).

## Appendix B. Non-implicitly Bayesian agents are vulnerable to adversarial bets

Consider a betting game in which two players — an agent and an adversary — make bets on the next sequence of $n$ observations, each observation taking value in some space $\mathcal{X}$. At the end, a transfer of funds happens between the agent and the adversary depending on the observed outcomes.

Formally, a *bet* is a function $r : \mathcal{X}^n \to \mathbb{R}$. The value of $r$ can be thought of as representing the transfer of funds from the adversary to the agent for each possible outcome; i.e. if the observed outcome is $x_1, \ldots, x_n$ and $r(x_1, \ldots, x_n)$ is positive, the agent receives $r(x_1, \ldots, x_n)$ from the adversary. If, on the other hand, the observed outcome is $x_1, \ldots, x_n$ and $r(x_1, \ldots, x_n)$ is negative, the agent gives $-r(x_1, \ldots, x_n)$ to the adversary.

We'll consider players that, in some sense, make predictions on the sequence of observations, and choose which bets to accept based on those beliefs. Concretely, *agent beliefs*, and *adversary beliefs* are probability mass functions on $\mathcal{X}^n$.

**Definition 6 (Bets favourable to agent)** *Given agent beliefs* $q : \mathbb{X}^n \to \mathbb{R}$*, a function* $r : \mathcal{X}^n \to \mathbb{R}$ *is said to be a **bet favourable to the agent** if:*

$$\sum_{x_{1:n} \in \mathcal{X}^n} r(x_{1:n}) q(x_{1:n}) > 0. \tag{9}$$

In other words, a bet is favourable to the agent if the 'expected return' from the point of view of the agent is strictly positive.

**Definition 7 (Bets favourable to adversary)** *Given adversary beliefs* $\bar{q} : \mathbb{X}^n \to \mathbb{R}$*, a function* $r : \mathcal{X}^n \to \mathbb{R}$ *is said to be a **bet favourable to the adversary** if:*

$$\sum_{x_{1:n} \in \mathcal{X}^n} (-r(x_{1:n})) \bar{q}(x_{1:n}). \tag{10}$$

**Definition 8 (Admissible bets)** *Given agent and adversary beliefs, a bet* $r$ *is said to be an **admissible bet** if it is simultaneously a bet favourable to the agent and a bet favourable to the adversary.*

**Definition 9 (Minimal bets)** *Given agent beliefs* $q : \mathbb{X}^n \to \mathbb{R}$*, and adversary beliefs* $\bar{q} : \mathbb{X}^n \to \mathbb{R}$*, a function* $r : \mathcal{X}^n \to \mathbb{R}$ *is said to be a **minimal bet** if it's a linear combination of agent and adversary beliefs, namely:*

$$r(x_{1:n}) = a q(x_{1:n}) + b \bar{q}(x_{1:n}) \qquad \qquad \textit{for some } a, b \in \mathbb{R} \tag{11}$$

The rationale for this is that adding components to the bet that are orthogonal to the beliefs of the agent and the adversary will not change the expected return of the bet for either of them. In other words, for any bet $r$ that is not minimal, there exists a 'smaller' minimal bet $r'$ such that the expected returns of $r$ and $r'$ are the same for both the agent and the adversary. Restricting ourselves to minimal bets hence simply ensures that the agents do not arbitrarily make bets that are not justified on the ground of their beliefs. This is made concrete in the remark below:

**Definition 10 ($\ell_2$ norm)** *For functions $f : \mathcal{X}^n \to \mathbb{R}$, the $\ell_2$ norm of $f$, written $\|f\|_2$, is defined as $\|f\|_2 = \sqrt{\sum_{x_{1:n} \in \mathcal{X}^n} f(x_{1:n})^2}$*

**Remark 11** *Given agent beliefs $q$ and adversary beliefs $\bar{q}$, and a minimal bet function $r : \mathcal{X}^n \to \mathbb{R}$, any other bet function $r' : \mathcal{X}^n \to \mathbb{R}$ for which:*

$$\sum_{x_{1:n} \in \mathcal{X}^n} r(x_{1:n}) q(x_{1:n}) = \sum_{x_{1:n} \in \mathcal{X}^n} r'(x_{1:n}) q(x_{1:n})$$
$$\sum_{x_{1:n} \in \mathcal{X}^n} (-r(x_{1:n})) \bar{q}(x_{1:n}) = \sum_{x_{1:n} \in \mathcal{X}^n} (-r'(x_{1:n})) \bar{q}(x_{1:n}),$$

*i.e. for which the expected returns to the agent and the adversary are the same as under $r$, will have a greater or equal $\ell_2$ norm: $\|r\|_2 \leq \|r'\|_2$, with equality if and only if $r = r'$.*

Given a bet function $r : \mathcal{X}^n \to \mathbb{R}$, and a *true distribution* $p : \mathcal{X}^n \to \mathbb{R}$, the game proceeds by sampling observations $X_1, \ldots, X_n$ from $p$ — i.e. $(X_1, \ldots, X_n) \sim p$ — and at the end of game a transfer of fund occurs. The *reward to the agent* is defined as $r\left((X_1, \ldots, X_n)\right)$, and the *expected return to the agent* is defined as:

$$\mathbb{E}[r\left((X_1, \ldots, X_n)\right)] = \sum_{x_{1:n} \in \mathcal{X}^n} p(x_{1:n}) r(x_{1:n}). \tag{12}$$

We'll specifically consider the case of an adversary that constructs an "exchangeable-ified" version of the agent's beliefs $q$ as:

$$\bar{q}(x_{1:n}) = \frac{1}{n!} \sum_{\pi \in \Pi_n} q(x_{\pi(1)}, \ldots, x_{\pi(n)}), \tag{13}$$

where $\Pi_n$ is the collection of all permutations of $n$ elements. It should be clear that $\bar{q}$ is itself exchangeable. One can think of interpret this as a scenario in which the adversary doesn't know the true probabilities of the events, but knows the beliefs of the agent, and so they are able to exploit a possible lack of exchangeability in the agent's beliefs.

The theorem we will state below roughly says "if a sensible bet is made between the agent and the adversary, then *no-matter what the true distribution is* the expected return for the agent is strictly negative". The minimality of the bet is required for this to hold: if the bets are allowed to not be minimal, then we could find some true distribution that rewards one of the bettors despite them having no 'edge' or advantage in terms of their beliefs.

**A note on notation** In what follows, it will be helpful to simplify the notation, and think of the agent and adversary beliefs, as well as bets, as vectors. If we enumerate all the elements of $\mathcal{X}^n$, we can represent functions from $\mathcal{X}^n$ to $\mathbb{R}$ as vectors in $\mathbb{R}^{|\mathcal{X}^n|}$. For example, if $\mathcal{X} = \{0, 1\}$, $n = 2$, and the chosen ordering of elements of $\mathcal{X}^n$ is $((0,0), (0,1), (1,0), (1,1))$, then the function $r$ can be represented as a vector in $\mathbb{R}^4$:

$$\boldsymbol{r} = \begin{bmatrix} r(0,0) \\ r(0,1) \\ r(1,0) \\ r(1,1) \end{bmatrix}$$

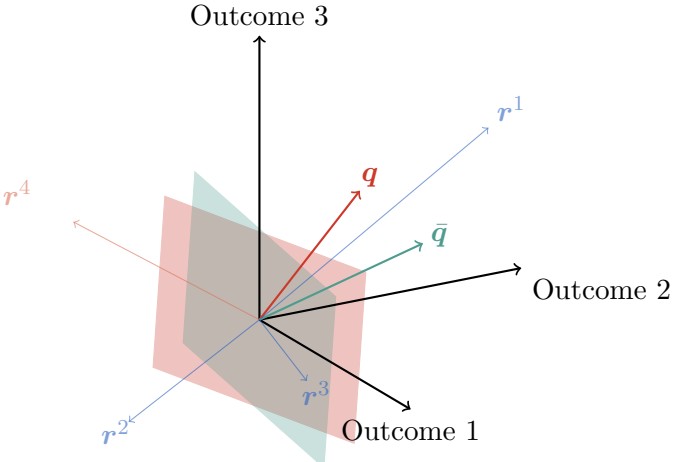

Figure 3: An illustration of the beliefs $q$ of the agent and the beliefs $\bar{q}$ of the adversary as vectors. The favourable bets for the agent are those that lie in the direction of $q$ of the shaded plane orthogonal to $q$. Similarly, the favourable bets for the adversary are those that lie in the direction of $-\bar{q}$ of the shaded plane orthogonal to $\bar{q}$. Multiple inadmissible bets $r_1, r_2, r_3$ are shown, as well as one admissible bet $r_4$.

for the reminder of this section (and this section only) we'll switch freely between the function notation and the vector notation for the same function as it should be apparent from the context which one is being considered.

This brings us to the following result, which we will now prove:

**Theorem 12** *Given an agent with beliefs $q : \mathcal{X}^n \to [0, 1]$, and an adversary with "exchangeable-ified" beliefs $\bar{q}$ as defined in (3), for any exchangeable true distribution $p : \mathcal{X}^n \to [0, 1]$ that has some shared common support with $q$ all minimal and admissible bets $r$ have strictly negative expected return for the agent:*

$$\boldsymbol{p} \cdot \boldsymbol{r} = \sum_{x_{1:n} \in \mathcal{X}^n} p(x_{1:n}) r(x_{1:n}) < 0 \tag{14}$$

*Furthermore, either:*

- *$q$ is exchangeable and $\bar{q} = q$, and there are no admissible bets.*

- *$q$ is not exchangeable and there exist admissible (and minimal) bets.*

**Remark 13** *In particular, Theorem 12 holds when $p$ is an i.i.d. distribution.*

**Proof**

For any exchangeable[7] function $f : \mathcal{X}^n \to \mathbb{R}$ we have that:

$$\boldsymbol{f} \cdot \bar{\boldsymbol{q}} = \sum_{x_{1:n} \in \mathcal{X}^n} f(x_{1:n}) \bar{q}(x_{1:n})$$

$$= \sum_{x_{1:n} \in \mathcal{X}^n} f(x_{1:n}) \left( \frac{1}{|\Pi_n|} \sum_{\pi \in \Pi_n} q(\pi(x_{1:n})) \right)$$

$$= \frac{1}{|\Pi_n|} \sum_{\pi \in \Pi_n} \sum_{x_{1:n} \in \mathcal{X}^n} f(x_{1:n}) q(\pi(x_{1:n}))$$

$$= \frac{1}{|\Pi_n|} \sum_{\pi \in \Pi_n} \sum_{x_{1:n} \in \mathcal{X}^n} f(\pi(x_{1:n})) q(\pi(x_{1:n}))$$

$$= \frac{1}{|\Pi_n|} \sum_{\pi \in \Pi_n} \sum_{x_{1:n} \in \mathcal{X}^n} f(x_{1:n}) q(x_{1:n})$$

$$= \sum_{x_{1:n} \in \mathcal{X}^n} f(x_{1:n}) q(x_{1:n}) = \boldsymbol{f} \cdot \boldsymbol{q},$$

where the second to last equality holds because permuting the elements of both vectors in a dot product $\boldsymbol{f} \cdot \boldsymbol{q}$ doesn't change the result. Hence:

$$\boldsymbol{p} \cdot \bar{\boldsymbol{q}} = \boldsymbol{p} \cdot \boldsymbol{q} \qquad\qquad \bar{\boldsymbol{q}} \cdot \bar{\boldsymbol{q}} = \boldsymbol{q} \cdot \bar{\boldsymbol{q}} \qquad\qquad (15)$$

**I. All admissible and minimal bets have negative expected return.** Suppose $r$ is minimal and admissible. Then, by minimality, $\boldsymbol{r}$ can be represented as:

$$\boldsymbol{r} = a\boldsymbol{q} + b\bar{\boldsymbol{q}}, \qquad\qquad (16)$$

for some constants $a, b \in \mathbb{R}$. By admissibility $r$, we can further get the following inequality for $a, b$:

$$\boldsymbol{q} \cdot \boldsymbol{r} > 0 \qquad \Rightarrow a(\boldsymbol{q} \cdot \boldsymbol{q}) + b(\boldsymbol{q} \cdot \bar{\boldsymbol{q}}) \qquad\qquad\qquad > 0 \qquad\qquad (17)$$

$$\bar{\boldsymbol{q}} \cdot \boldsymbol{r} > 0 \qquad \Rightarrow a(\bar{\boldsymbol{q}} \cdot \boldsymbol{q}) + b(\bar{\boldsymbol{q}} \cdot \bar{\boldsymbol{q}}) = a(\bar{\boldsymbol{q}} \cdot \boldsymbol{q}) + b(\bar{\boldsymbol{q}} \cdot \boldsymbol{q}) \qquad < 0$$

$$\Rightarrow a + b \qquad\qquad\qquad\qquad < 0 \qquad , \qquad (18)$$

as $\boldsymbol{q} \cdot \bar{\boldsymbol{q}} > 0$.

By (16), we can rewrite the expected return (for the agent) as:

$$\boldsymbol{p} \cdot \boldsymbol{r} = a\boldsymbol{p} \cdot \boldsymbol{q} + b\boldsymbol{p} \cdot \bar{\boldsymbol{q}} = a\boldsymbol{p} \cdot \boldsymbol{q} + b\boldsymbol{p} \cdot \boldsymbol{q} = (a + b)(\boldsymbol{p} \cdot \boldsymbol{q}),$$

where the second to last equality follows by (15).

Now, $\boldsymbol{p} \cdot \boldsymbol{q} > 0$ as long as $\boldsymbol{p}$ and $\boldsymbol{q}$ have some common support, and by (18), $a + b < 0$, hence:

$$\boldsymbol{p} \cdot \boldsymbol{r} < 0,$$

as required.

---

7. Exchangeable here has been appropriated in the context of general functions to mean that the function $f : \mathcal{X}^e \to \mathbb{R}$ is permutation invariant in its arguments, i.e. for any permutation $\pi : \{1, \ldots, n\} \to \{1, \ldots, n\}$, $f(x_{1:n}) = f(\pi(x_{1:n}))$.

**II. Minimal and admissible bets exist if and only if $q \neq \bar{q}$.** Suppose that $q = \bar{q}$. An admissible bet must satisfy:

$$0 < r \cdot q$$
$$0 > r \cdot \bar{q} = r \cdot q,$$

which is a contradiction, and so $q = \bar{q}$ implies that no admissible bet exists.

Now, suppose that $q \neq \bar{q}$. Note that:

$$\|\bar{q}\|_2^2 = \sum_{x_{1:n} \in \mathcal{X}^n} \left( \frac{1}{|\Pi_n|} \sum_{\pi \in \Pi_n} q(\pi(x_{1:n})) \right)^2$$

$$\leq \sum_{x_{1:n} \in \mathcal{X}^n} \frac{1}{|\Pi_n|} \sum_{\pi \in \Pi_n} q(\pi(x_{1:n}))^2 \qquad \triangle \text{ GM-HM inequality}$$

$$= \sum_{x_{1:n} \in \mathcal{X}^n} q(x_{1:n})^2 = \|q\|_2^2,$$

and so, $q \cdot q > q \cdot \bar{q}$ (strict inequality, as $q \neq \bar{q}$).

Hence, picking $a = \left( \frac{q \cdot \bar{q}}{q \cdot q} \right)^{\frac{1}{2}}$ and $b = -1$ yields an admissible and minimal bet $r = aq + b\bar{q}$, because:

$$a(q \cdot q) + b(q \cdot \bar{q}) = (q \cdot \bar{q})^{\frac{1}{2}} \overbrace{(q \cdot q)^{\frac{1}{2}}}^{>(q \cdot \bar{q})^{\frac{1}{2}}} - (q \cdot \bar{q}) > 0$$

$$a + b = \underbrace{\left( \frac{q \cdot \bar{q}}{q \cdot q} \right)^{\frac{1}{2}} - 1}_{<1} < 0.$$

Hence, whenever $q \neq \bar{q}$ we can find a minimal and admissible bet as required.

∎

In the context of prediction rules, one can see the above result as a failing of not implicitly Bayesian prediction rules; such prediction rules would induce non-exchangeable distributions over finite sequences of observations (for some sequence length), and hence an agent that derives their beliefs from such a prediction rule would be vulnerable to the adversarial construction in Theorem 12.

## Appendix C. Bayesian Linear Model Prediction Rule

$$s_n \left( \begin{bmatrix} x_{n+1} \\ y_{n+1} \end{bmatrix} \mid \begin{bmatrix} x_1 \\ y_1 \end{bmatrix}, \dots, \begin{bmatrix} x_n \\ y_n \end{bmatrix} \right) = \mathcal{N} \left( y_{n+1}; \bar{\mu}, \bar{\Sigma} \right)$$

$$\bar{\mu} = \left( x_{n+1} \Phi^\mathsf{T} + \sigma^2 \right) \left( \Phi \Phi^\mathsf{T} + \sigma^2 \right)^{-1} y_{1:n}$$

$$\bar{\Sigma} = x_{n+1}^\mathsf{T} x_{n+1} - \left( x_n \Phi^\mathsf{T} + \sigma^2 \right) \left( \Phi \Phi^\mathsf{T} + \sigma^2 I \right)^{-1} \left( \Phi x_{n+1} + \sigma^2 \right) + \sigma^2$$

with $\Phi = \begin{bmatrix} x_1 & \dots & x_n \end{bmatrix}^\mathsf{T}$ and $y_{1:n} = \begin{bmatrix} y_1 & \dots & y_n \end{bmatrix}^\mathsf{T}$.

# Appendix D. Additional results

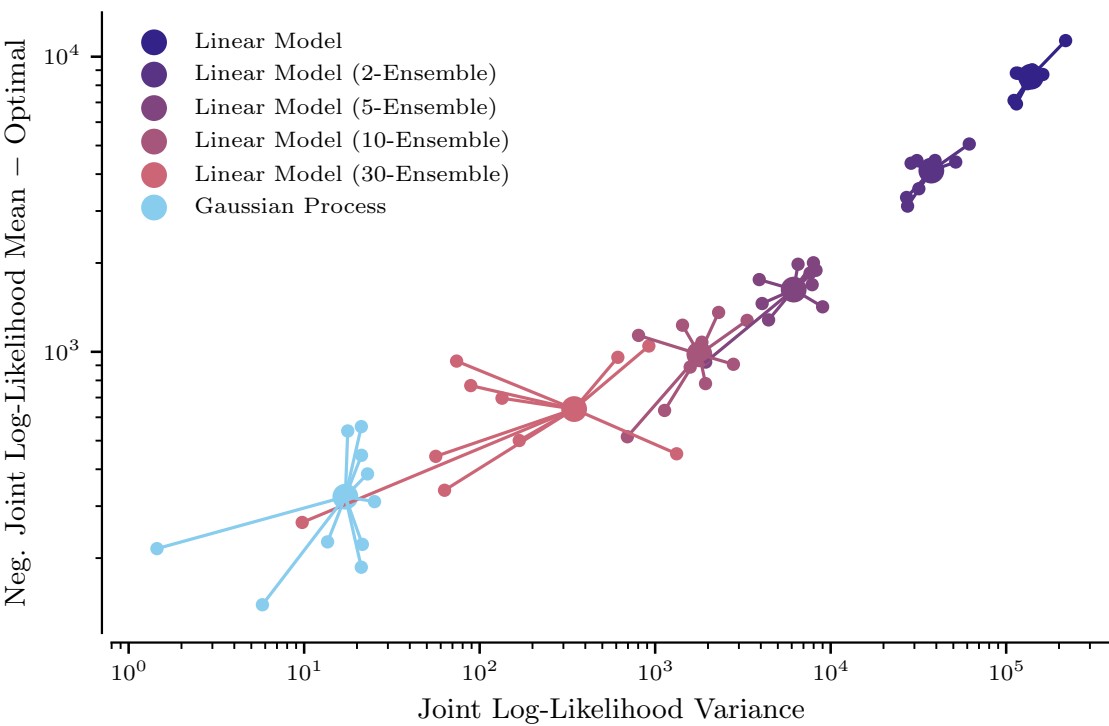

Figure 4: Joint log-likelihood variance as a measure of *implicit Bayesianness* (defined in (5)) vs. the negative sum of log-likelihood as a measure of *performance* (defined in (6)) plot for a Gaussian Process with a squared exponential kernel, and a linear model fit with the prior networks procedure on the regression task described in Section 4.2. The individual small dots depict the result for a particular collection of datapoints $((x_1, y_1), \ldots, (x_{100}, y_{100}))$ drawn from the data-generating distribution, and the large dots represent the mean over multiple dataset samples from the data-generating distribution. The expected negative log-likelihood of the *optimal* predictions on this dataset (i.e. using the true distribution) is subtracted from the mean negative joint log-likelihood plotted.

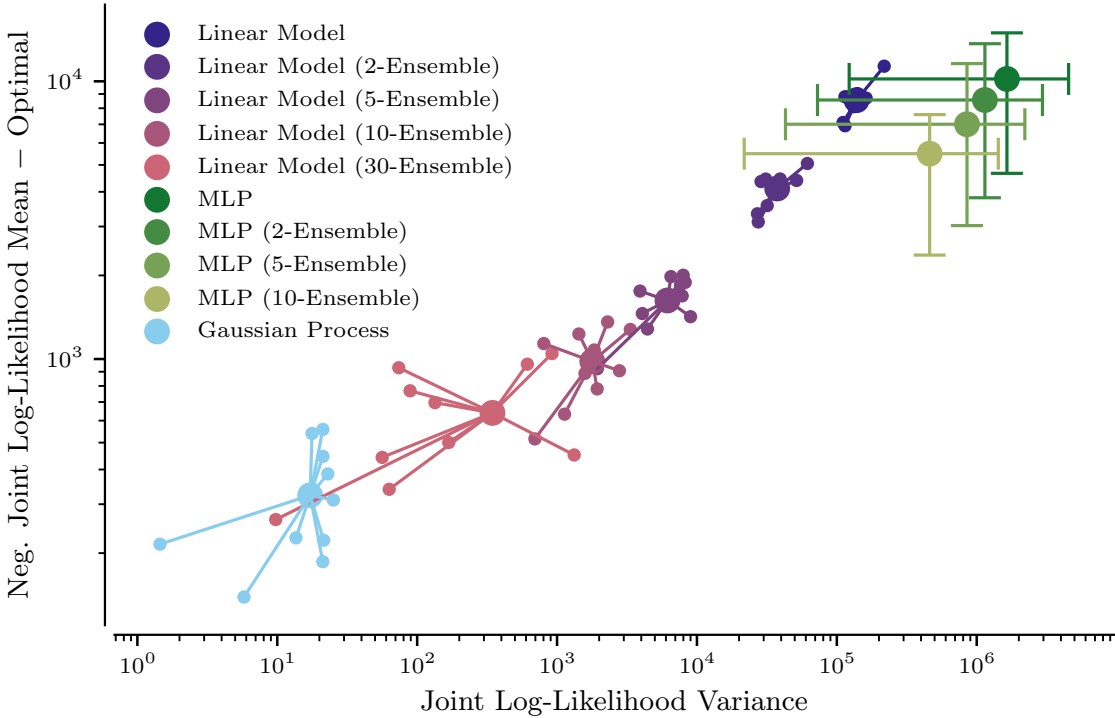

Figure 5: Comparison of the MLP results in Figure 6 and a Gaussian Process Figure 4 on the same plot.

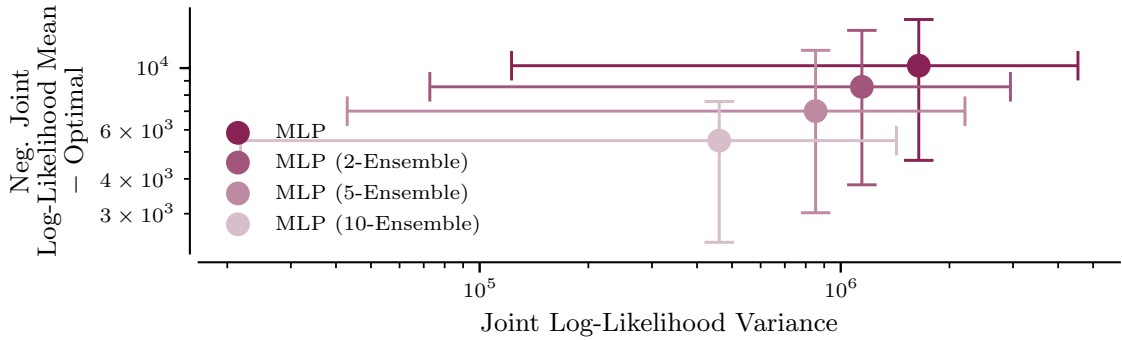

Figure 6: Joint log-likelihood variance as a measure of *implicit Bayesianness*(defined in (5)) vs. the negative sum of log-likelihood as a measure of *performance* (defined in (6)) plot for a MLP trained with SGD, as well as ensembles of such models, on the regression task described in Section 4.2. The error bars denote 10th and 90th percentiles over multiple dataset samples from the data-generating distribution.

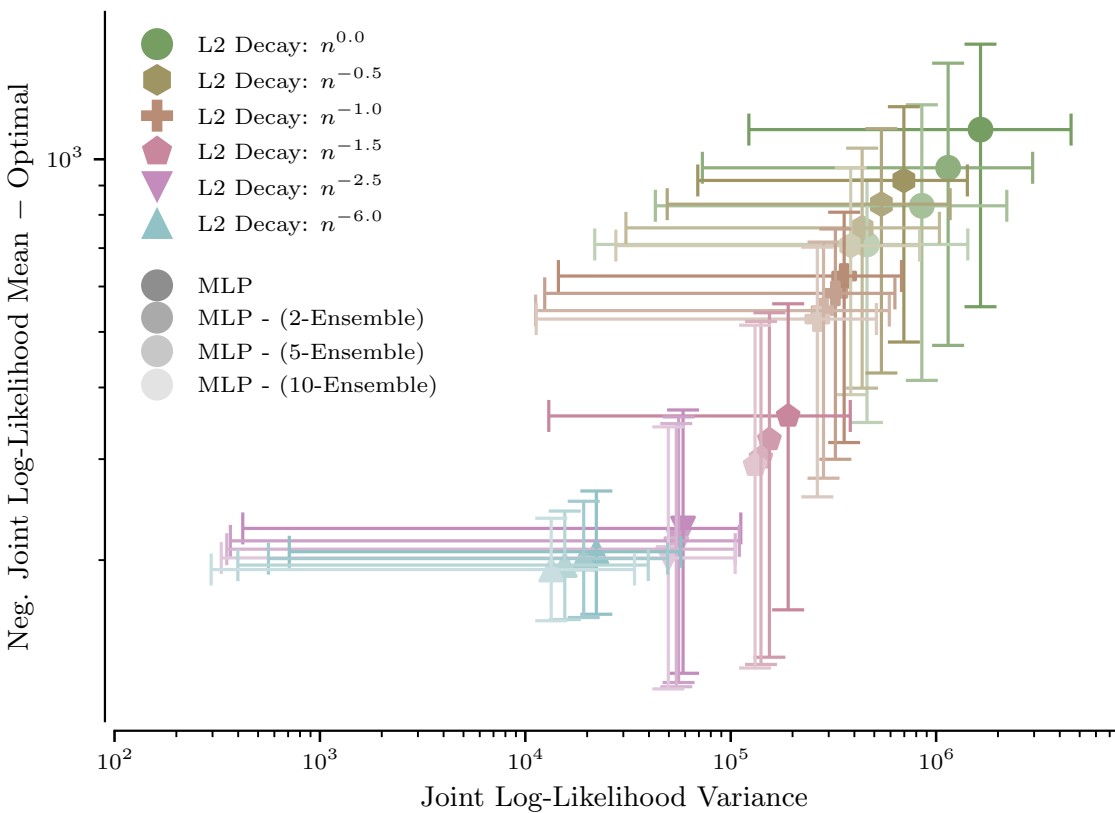

Figure 7: Comparison of different $\ell_2$-decay schedules (with respect ot dataset size) for an MLP model trained with SGD measuring joint log-likelihood variance as a measure of *implicit Bayesianness*(defined in (5)) vs. the negative sum of log-likelihood as a measure of *performance* (defined in (6)). Results shown for the regression task described in Section 4.2. The error bars denote 10th and 90th percentiles over multiple dataset samples from the data-generating distribution.

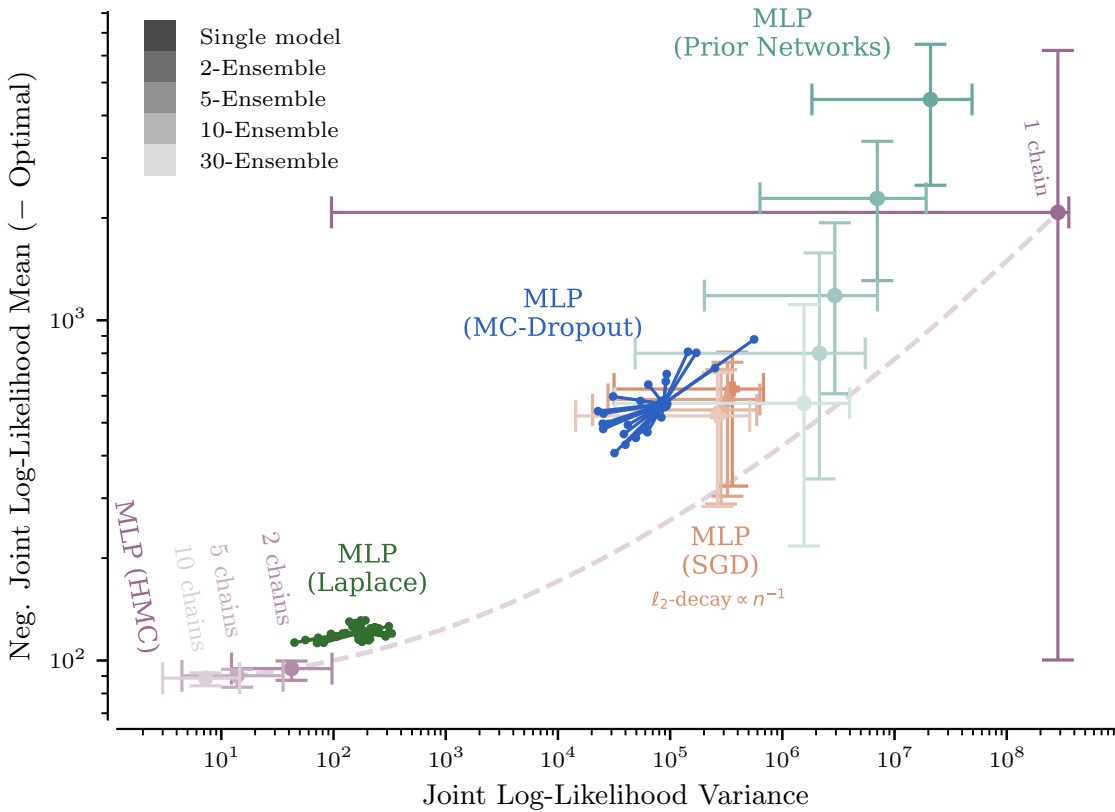

Figure 8: Joint log-likelihood variance as a measure of *implicit Bayesianness* (defined in (5)) vs. the negative sum of log-likelihood as a measure of *performance* (defined in (6)) plot comparing various approximate methods for inference in a 3-hidden layer Bayesian Neural Network. In particular, the performance of the prior network method (Osband et al., 2018) is subpar even compared to regular SGD training of the neural network in both performance and implicit Bayesianness.

## D.1. Where does the performance advantage of different models come from?

The results in Figure 2 illustrate that Multi-layer Perceptrons (MLPs) and MLP ensembles trained with SGD appear to have subpar performance compared to other methods, especially GPs, in terms of the joint log-likelihood. This might be somewhat surprising, especially given the model mismatch of the true function for the GP models. We might expect that the MLP models should eventually outperform GPs if the training size was large enough. In this subsection, we break down the performance metric in Figure 2 to try and gauge what's going on.

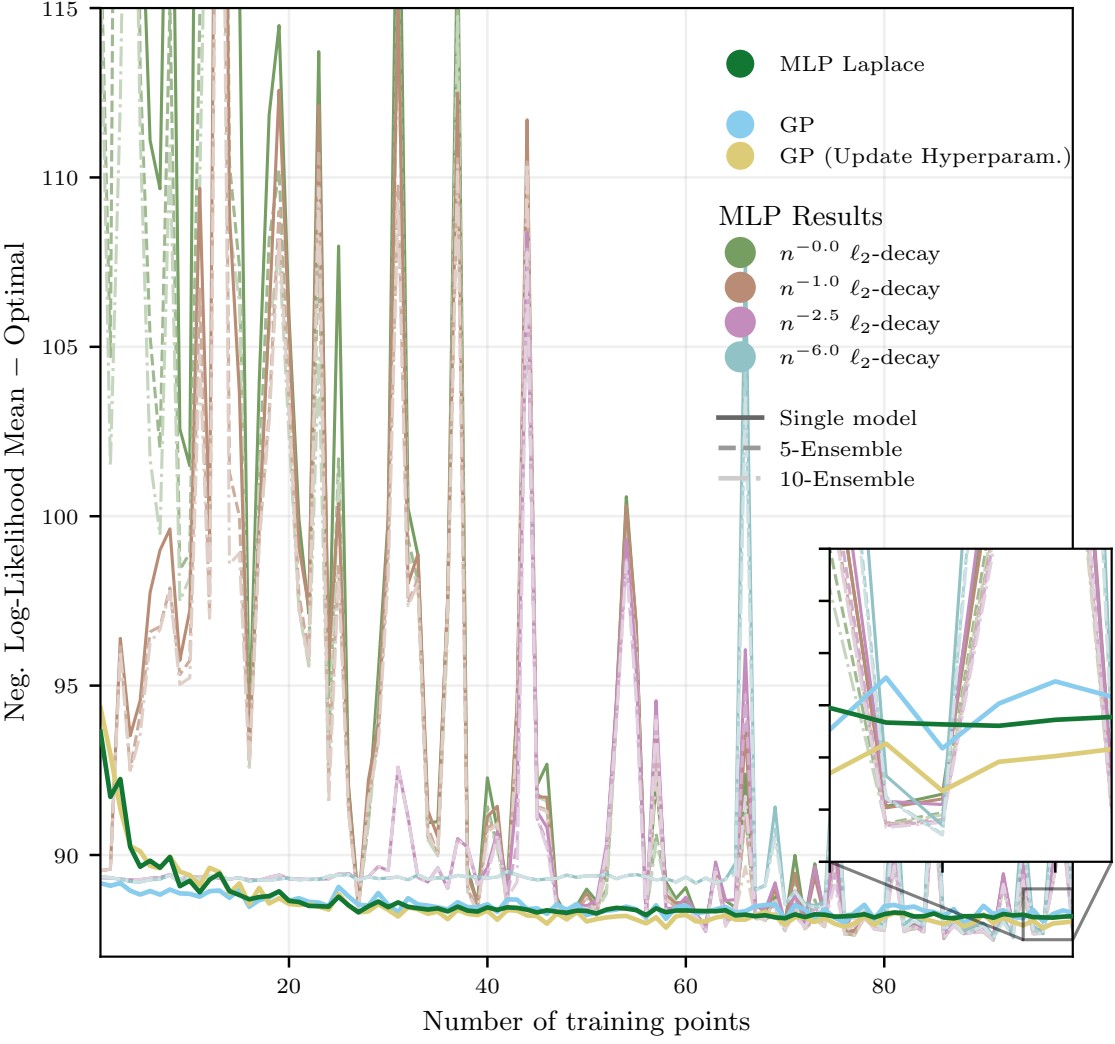

Figure 9: Comparison of the average "test" log-likelihoods of the next datapoint in a sequence for the different models with different training data sizes (number of conditioning datapoints).

In Figure 9, we plot the average log-likelihoods for the different methods broken down by the training set size. The final 'joint' log-likelihood metric in Figure 2 will be the sum of the per-training-size log-likelihoods. The figure illustrates that the SGD-trained MLP models can overfit dramatically when trained with very few training datapoints ($< 40$). This seems to be the biggest contributor to their subpar aggregate performance. Surprisingly, for the standard $\ell_2$-decay of $n^{-1}$ (with $n$ being the training data size), the performance doesn't seem to monotonically improve, and is actually the worst after observing around 15 to 20 datapoints.

Increasing the regularisation strength through $\ell_2$-decay seems to have the effect of averting the worst of the overfitting in MLPs in the low-data regime by essentially forcing the network to ignore the training data until a certain training dataset size is reached. For example, predictions from an MLP with $n^{-2.5}$ $\ell_2$-decay are shown in Figure 10.

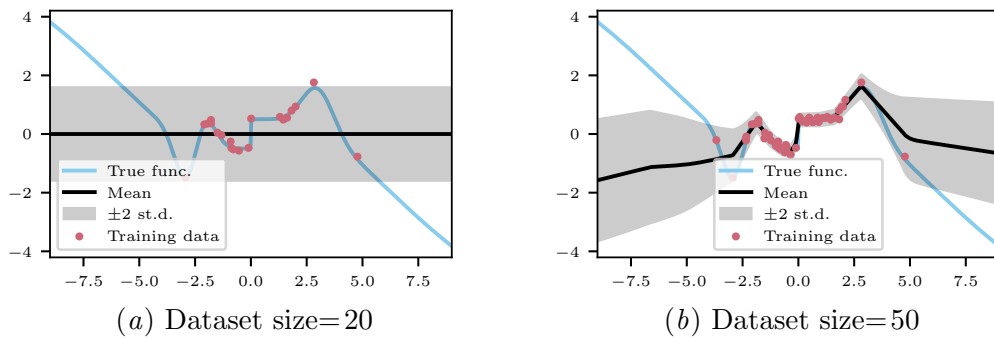

$(a)$ Dataset size$=20$        $(b)$ Dataset size$=50$

Figure 10: MLP trained with SGD on the discontinuous regression task with $\ell_2$-decay of $n^{-2.5}$ for different training data sizes.

Ensembling the MLP predictions seems to help surprisingly little with performance in the low-data regime. This appears to be because, for some training data configurations, the MLPs trained with different seeds overfit in the same way. An example of this is shown in Figure 11.

It appears that both MLPs fit with the Laplace approximation and a GP with kernel hyperparameters updated throughout training eventually outperform the GP with fixed hyperparameters. This is at the cost of the initially subpar performance in the low-data regime ($< 20$ datapoints), where updating the kernel hyperparameters in the GP might lead to overfitting.

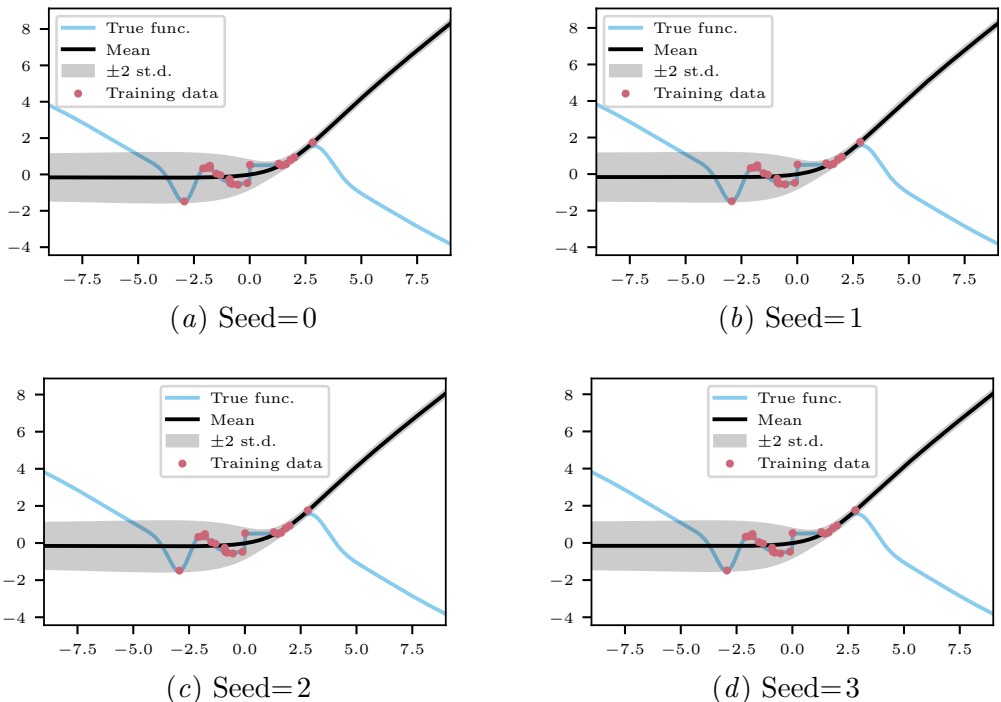

(a) Seed=0 (b) Seed=1

(c) Seed=2 (d) Seed=3

Figure 11: MLP trained with SGD on the same 19 datapoints on the discontinuous regression task with different seeds (different initialisation). The model consistently fits the data in the same manner, regardless of initialisation, extrapolating the data unfavourably in the interval $(-2.5, \infty)$.

## Appendix E. Can a prediction rule at a fixed step be extended to an implicitly Bayesian prediction rule?

In deep learning, we would often tailor our approach to the amount of data available at hand. For example, we might use a small neural network for a small dataset, and a large neural network for a large dataset. If we have fewer than hundreds of datapoints, we might not even consider using a neural network at all. Hence, viewing training of a neural network as a prediction rule for all possible data sizes might not be the most natural lens.

One might wonder, assuming that one has only specified a prediction rule at step $n$ only, can it be extended to a prediction rule for all $n$ to satisfy some of the previously mentioned properties?

One reasonable guess might be that, if the prediction rule at step $n$ is invariant to the ordering of the data, i.e. $s_n(\cdot|x_1, \ldots, x_n) = s_n(\cdot|x_{\pi(1)}, \ldots, x_{\pi(n)})$ for any permutation $\pi$ of $\{1, \ldots, n\}$[8], then maybe it can be extended to an exchageable prediction rule. If that was the case, that would be good news: as long as we're making the prediction at step $n$ only, we can claim that our prediction rule is implicitly Bayesian; once we start making predictions at other time-steps, we just need to figure out what an implicitly Bayesian extension is.

---

8. This is a weaker condition than exchangeability, as it only requires invariance to the ordering of the conditioning sequence.

However, the invariance of the prediction rule at step $n$ to the ordering of the data is not a sufficient condition for an implicitly Bayesian extension to exist. It's possible to devise simple counter-examples showing to the contrary.

# Appendix F. Experimental Details

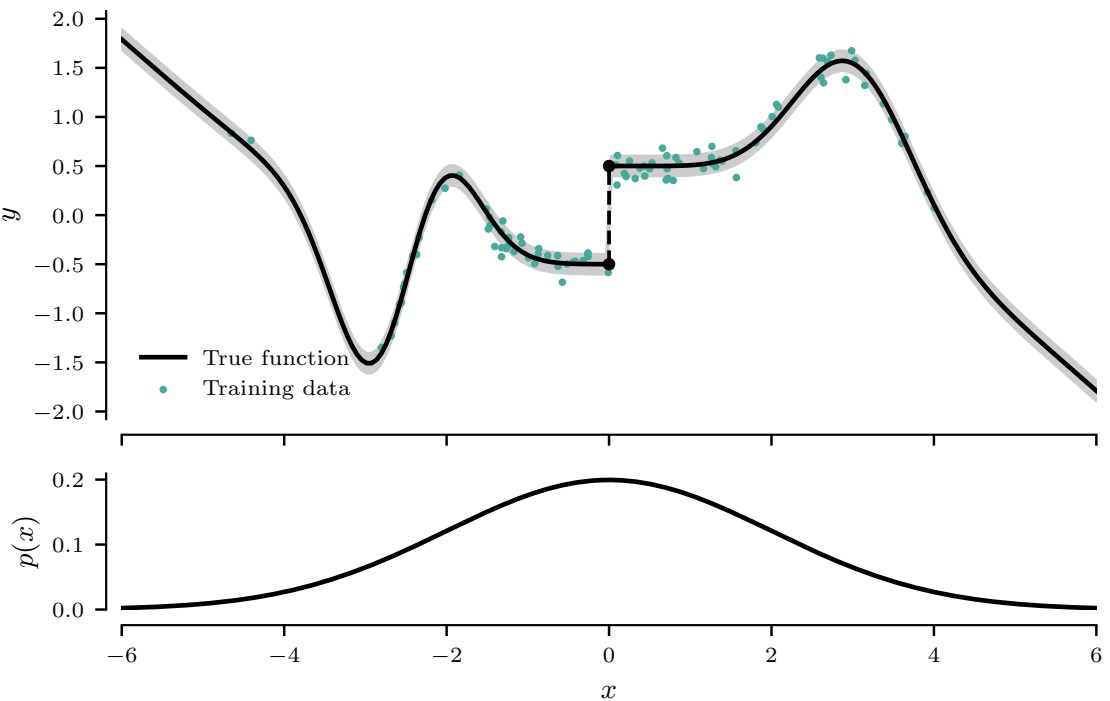

Figure 12: A 1D regression task. *(Bottom)*: The Gaussian density on the covariates. *(Top)*: The data-generating process illustrating the true function, with the filled area around it depicting the standard deviation of the homogeneous Gaussian noise.

## F.1. Gaussian Process and Prior Networks

For the Gaussian Process, we consider a squared exponential kernel of the form $k_{\mathrm{SE}}(x, x') = a^2 \exp\left(-\frac{(x-x')^2}{2\ell^2}\right)$. By default, the output variance $a^2$ and the length-scale $\ell$ are both set to 1. When the hyperparameters are being updated using margina likelihood, we set the ranges for both $a^2$ and $\ell$ to $(10^{-5}, 10^5)$ and optimise using LBFGS.

For the feature expansion for the Prior Network, we use a large number of Gaussian radial basis function features uniformly spaced over the relevant part of the input domain; in the limit, if the basis functions are adequately chosen, this should yield a Bayesian Linear Model equivalent to a squared exponential kernel GP Rasmussen and Williams (2005). Concretely, we use a Gaussian basis function $\phi_\beta(x) = c \exp(-\frac{(x-\beta)^2}{\ell^2})$ for an adequately chosen constant $c$ ($\approx 4.47$) with 100 bases spaced equidistantly on the interval $[-6.0, 6.0]$.

The variance of the noise, both for the likelihood of the Gaussian Process and the linear model, were in both cases set to the true (known) variance for the task.

### F.2. Ensembling

Models are ensembled by averaging their predictions: $\frac{1}{M} \sum_{j=1}^{M} s_i^{\epsilon_j}(x^*|x_1,\ldots,x_i)$ (equivalent to taking the `logmeanexp` operation of the log-probabilities for test data).

### F.3. MLP

The hyperparameters of the model (weight decay, learning rate) were tuned on a validation set, with a separate randomly sampled training set of size $n$, and fixed for all steps $i \leq n$ of the prediction rule $s_i$ (except where explicitly stated that they were adjusted). The hyperparameters found with random search for the 1D regression task in Figure 12 were a learning rate of $10^{-3}$ and weight-decay of 0.017. In Figure 13 we show a grid-search over the weight-decay to validate 0.017 is close to optimal.

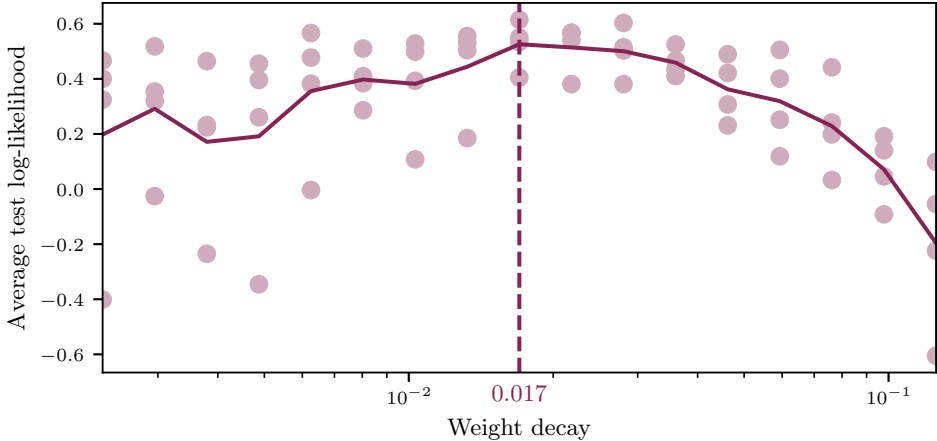

Figure 13: Grid-search over weight-decay for the MLP experiments on the 1D regression task.

The neural network model considered in regression tasks is an MLP with three hidden layers, each of width 512, with a `ReLU` activation and no normalisation. The model was trained for 10000 iterations (independently of dataset size) with a batch-size of 64 (or less, if the dataset size is smaller).

**Heteroscedastic vs. homoscedastic noise output** On a regression task, we could parameterise a Gaussian likelihood with a neural network by either **a)** only parameterising the mean with the output of the neural network, and fixing the scale to some value (e.g. the true noise standard deviation), or by **b)** parameterising both the mean and the scale with the outputs of the neural network. For a single modelto be able to represent epistemic (knowledge) uncertainty (with a non-trivial prior), however, it is necessary to be able to vary the output uncertainty depending on the input. Hence, wherever possible, we use the heteroscedastic likelihood (option **b)**) wherever this doesn't break the theoretical justification for a given method. Concretely, we use the heteroscedastic noise likelihood (with scale parameterised by a `softplus` transformation of the network output) for all MLP experiments with SGD and SGD ensembles, all $\ell_2$ decay experiments, MC-dropout (as the variational approximation is compatible with heteroscedastic nosie output), and HMC. The

Laplace method and prior networks by default assume a fixed homoscedastic noise likelihood, and hence that's the likelihood we used for those experiments (with the true noise standard deviation for the scale).

### F.3.1. $\ell_2$-DECAY EXPERIMENTS

For different $\ell_2$-decay experiments, we run the same experiments with same hyperparameters as above, but with a decay schedule in the dataset size $i$ of the form $c_\alpha i^{-\alpha}$ for different values of $\alpha$. $c_\alpha$ is in each case set so that the value of $\ell_2$ decay would match for $i = n$, i.e. the final value of the weight-decay would be the optimal one as found with cross-validation.

For larger (negative) values of $\alpha$, the initial weight-decay for small dataset sizes gets so large that the training becomes numerically unstable. Hence, we clip the maximum value of weight-decay to 2000, which is sufficient for the model to always learn the solution equivalent to all weights being set to 0.

For all results for all methods, for each dataset sequence $((x_1, y_1), \ldots, (x_n, y_n))$, we ignore the prediction for the first datapoint in the sequence $(x_1, y_1)$, effectively setting the log-probability assigned by $s_0$ to 0 for all methods. This does contribute a small amount to increase in the variance of the log-joint even for exactly exchangeable methods. We do so as, for deep learning methods, the prediction with no training data is somewhat ill-defined.

### F.3.2. LAPLACE APPROXIMATION

For the Laplace approximation (Ritter et al., 2018; MacKay, 1992), we use post-hoc Laplace after training the network with SGD. As the Laplace approximation traditionally expects an optimised set of parameters corresponding to the maximum-a-posteriori estimate, we set the weight decay to match the prior precision in the Laplace approximation; in other words, if the prior precision is $\lambda$, weight decay will be set to $\frac{\lambda}{n}$ for a training dataset of size $n$. We don't optimise prior-precision post-hoc during each training run (as choosing the prior with emprical Bayes is likely to make the model less implicitly Bayesian). We use the same number of optimisation steps as for regular MLP training for finding the MAP solution. For the Laplace approximation, we use a Kronecker-factored (K-FAC) approximation for the Hessian, as described in (Daxberger et al., 2022). The predictive distribution is the Generalised Linear Model predictive, which for regression with a Gaussian likelihood is available in closed form.

One difference to other MLP experiments is that, to be compatible with standard implementations of Laplace, the MLP parameterises only the mean of a homoscedastic likelihood (where the standard deviation in the likelihood function is set to the true data-generating process noise standard deviation). This is in contrast to other MLP experiments, where the MLP parameterises both the mean and the standard deviation in the likelihood.

We again ablate over the learning rate and prior precision and pick the parameters that yield highest validation-set log-likelihood, as shown in Figure 14.

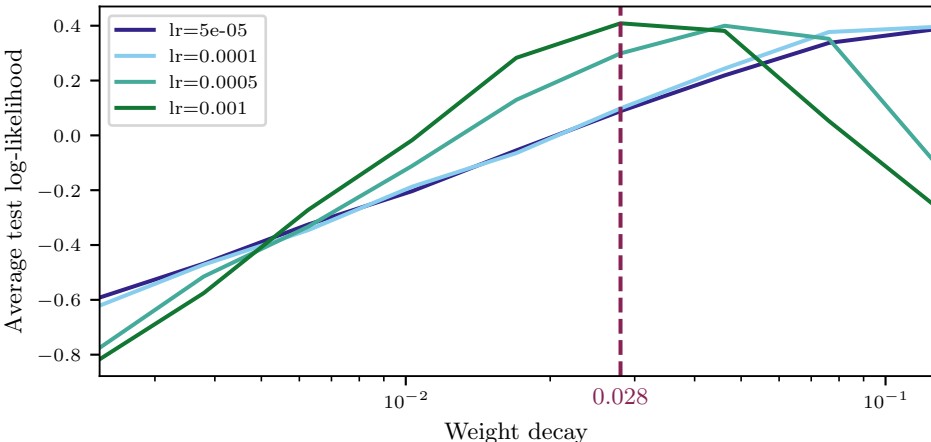

Figure 14: Grid-search over weight-decay and learning rate for the MLP experiments with the **Laplace approximation** on the 1D regression task. Weight-decay is equal to prior precision divided by dataset size (100).

