# OpenReview forum: "Implicitly Bayesian Prediction Rules in Deep Learning"
_approximateinference.org/AABI/2024/Symposium_Archival_Track — AABI 2024 - Archival Track_

### Official Review · Reviewer_XqiK · 2024-04-08
**interesting work**

**Rating:** 7
**Confidence:** 4

**Review:**

This work proposes to evaluate the "Bayesianness" of arbitrary prediction rules.  The authors discussed equivalent characterisations for a prediction rule (of exchangeable or iid data) being Bayesian, and proposed a practical measure based on the variance of cumulative log losses computed on randomly permuted observations.  Experiments on a toy regression problem compared the Bayesianness of the perturbed ensemble method of Osband et al (2018) with that of GP and regularized MLP predictors.

The proposal to measure the implicit Bayesianness of commonly used prediction rules is timely given the common belief that various apparently non-Bayesian methods are "implicitly Bayesian". Ultimately it would be very interesting if we could apply appropriately chosen methods to compare the the "implicit Bayesianness" of non-Bayesian ensemble methods with that of explicitly Bayesian methods for neural network models, as the convergence of the latter is hard to guarantee in practice.  The current results that are, in my view, sufficiently interesting to justify their presentation at the symposium.  Below I provide a few detailed questions/comments.

1. Exact Bayesian approaches enjoy various benefits; for instance we can extract epistemic uncertainty estimates (e.g., Kendall & Gal, 2017), or validate a certain model by validating the appropriateness of its prior and likelihood separately. An ideal measure of "implicit Bayesianness" should assign higher ratings to prediction rules that preserve more benefits of exact Bayesian rules. Can we say this about the proposed metric, or is it possible to derive metrics that are closer to this ideal?

2. The proposal to measure the implicit Bayesianness through the variance of log predictive densities is reasonable if we compare two prediction rules with a roughly comparable predictive power. But is it still reasonable if we compare two rules with significantly different values for the expected log loss?  This is relevant for Fig.2 where the expected log loss incurred by MLPs with different regularization scalings are very different, so it could be an alternative explanation to "non-Bayesian regularization scalings are more 'implicitly Bayesian'".

3. The cumulative log density in Eq.(5) is often dominated by the "earlier values" corresponding to smaller choices of $i$. But we are not generally interested in the predictive performance for very small values of $i$. Is there any reason not to start the summation from some $i\ge n_0\gg 1$, or how will the experimental results change if we change the measure in this way?

4. How does the method of Osband et al (2018) perform when applied to MLPs?  It would also be interesting to apply the proposed measure to SGLD and HMC on the same MLP models.

5. What are the potential practical implications we may derive from results like Theorem 3?


### References

Kendall, Alex, and Yarin Gal. "What uncertainties do we need in bayesian deep learning for computer vision?." Advances in neural information processing systems 30 (2017).

---

### Official Review · Reviewer_Y6nz · 2024-04-22
**A section of an interesting line of work**

**Rating:** 6
**Confidence:** 3

**Review:**

## Summary

This paper brings the concept of implicit Bayesianness to the evaluation of non-Bayesian models, especially deep learning models. The conditions for being implicit Bayesian are constructed (exchangeability, stationary&c.i.d., spreadability). A metric based on the variance of the log joint density with all permutations of data is proposed. The metric is estimated by Monte Carlo sampling of permutations and random seeds. Implicit Bayesianness vs performance is tested on a 1D regression task with Gaussian processes, ensemble methods and MLPs. The vanilla GP is indeed quite Bayesian, but can trade some Bayesianness for performance by optimizing kernel hyperparameters. The number of ensembles and the scale of regularization are two factors that are expected to influence the Bayesianness of a method. Experiment results that reflect on them are provided.

Overall, I feel it will be nice to present this work at AABI.

## Positives

- There have been many advocates for generalized Bayesian inference which directly works with the prediction. One approach is to develop advanced Bayesian methods that calibrates well for prediction. The other approach, which is what this work takes, is to validates or incorporates Bayesianness in non-Bayesian methods. The proposed metric is quite novel without many assumptions. I feel it should be possible to generalize it to more modern deep learning methods. See the questions sections below.
- Though the model considered in the experiments is small, extensive experiments are performed and presented in Figure 2. The results from the experiments are intuitive and inspiring, which demonstrate the trade-offs between performance and implicit Bayesianness that a model may take.

## Negatives

- I feel the presentation of Section 3 could be improved. I understand it might be better to have less formulas for general audience, but I have a hard time understanding many terminologies in this section such as "event", "belief" and "bet". It would be better to have a part that clearly defines the probability spaces and the game states.
- The complexity of the method should be included. The chain rule of probability on the entire dataset is considered. If I am understanding correctly, it implies that for a single permutation, the model will be retrained $N$ times, where $N$ is the dataset size. The proposed metric is novel and exploratory, which will be improved in follow-up works. The complexity would be a good baseline for those works to refert to and compare with.

## Questions

- Many modern deep networks are overparameterized for the problem/data. I assume the implicit Bayesianness check will work with even fewer data, such that it becomes even more overparameterized. Is it possible to incorporate some theories from that field to predict the implicit Bayesianness given results on fewer data?
- I would like to know about the relation between the proposed metric vs cross-validation. After reading the work, I view cross-validation as validating the implicit Bayesianness for a single $s_i$.

---

### Official Review · Reviewer_MPvh · 2024-04-23

**Rating:** 5
**Confidence:** 4

**Review:**

This paper advocates for the benefits of implicit Bayesian predictions in the context of machine learning. Specifically, they include coherent updates of predictions under new data and the use of the Dutch book argument alongside with vulnerability to adversarial attacks of non-implicit Bayesian rules. In spite of these advantages, Bayesian methods often perform worse in several machine learning tasks. Thus, the paper proposes a method to measure how close a method is to performing similar to a Bayesian principle. For this, several properties of Bayesian prediction rules are described among which exhangability is chosen to be measured in practical applications. Exchangability is measured simply as the variability of the joint distribution under different factorizations given by random permutations of the observed variables. The paper then compares several methods in terms of this metric and also in terms of prediction performance, showing that pure Bayesian methods often perform better in terms of exchangability. The authors also show that some strategies such as ensembling not only improved prediction performance but also implicit Bayesianness.

Overall I believe this is an interesting paper. However, the methods proposed to measure implicit Bayesianness are rather simple and, although they do provide some insight about the benefits of some principles such as ensembling or weight regularization, they do not provide an appealing strategy or method to improve current deep learning methods at all. Therefore I believe that the overall contribution of the paper is small. I therefore still believe that this paper requires more work by the authors before it can be included in the proceedings track.

---

### Official Review · Reviewer_bLoo · 2024-04-26
**Interesting paper, but focus could be shifted.**

**Rating:** 5
**Confidence:** 4

**Review:**

**Summary:** This paper argues that focusing on making prediction rules Bayesian might be a fruitful route for deep learning instead of placing a prior over parameters. They then recap equivalent properties that are required for making a prediction rule Bayesian. The authors then prove that agents whose prediction rules are not Bayesian are vulnerable to adversarial bets. This leads to measures for how Bayesian a prediction rule is, along with a measure of how good the prediction rule is. Finally the authors then use these measures on Gaussian processes as well as various attempts at Bayesian NNs.

**Strengths:**
- The idea is interesting and seems like a novel way of thinking about how to incorporate Bayesian principles in models like Neural networks. The extent to which this is done in the paper is not clear (see below).
- The writing is clear.
- Measuring how Bayesian certain prediction rules are is a valid contribution, especially when considering approximate Bayesian methods with Neural networks. It would have been nice to see more Bayesian Neural network methods in the results to see how Bayesian their implied prediction rules actually are.

**Weaknesses:**
The biggest weakness of this work is that despite the setup, the results and recommendations in section 4 are a bit underwhelming. The paper argues that they present a different perspective on how to incorporate Bayesian principles into deep learning, however they end up measuring the 'Bayesianness' of prediction rules of existing approximations to Bayesian methods. There don't seem to be any recommendations of how one would use their insights to actually construct methods that are Bayesian. This is fine in itself if it wasn't hinted at in the rest of the paper and in the conclusion. The contribution of this paper can simply be the theorem and the measurement of Bayesian prediction, which could have then included measurements of the Bayesianness of multiple Bayesian NN approximations (ensembling, VI, Laplace etc.).

**Questions:** There are a few things in the results that seem a bit odd.
- What is the "Joint" in the x and y axis in Figure 2?
- The paper claims that the true functions was chosen to yield a model mismatch for a GP with a smooth kernel. What is the motivation behind this?
- Despite the claim above, it is surprising that the GP has better performance (lower log-like) than MLPs (Figure 5). Do you know what lead to this?
- Could you explain how the insights in the paper can actually be used to construct Neural networks with implicit Bayesian prediction rules? It is claimed in Section 4 "... show that simple design choices can lead to more or less implicitly Bayesian prediction rules", but where is this shown? Methods shown in section 4.2 are *known* to be approximations to Bayesian methods, so how does the insights from the rest of the paper help here?
- Section 2.3 is very interesting but it seems that only Definition 1 is used in the rest of the paper, am I correct in thinking that?

---

### Official Review · Reviewer_cHUM · 2024-05-01
**Implicitly Bayesian Prediction Rules in Deep Learning**

**Rating:** 6
**Confidence:** 3

**Review:**

The manuscript is generally written with the intention to be careful and the topic is thought provoking, as are recent papers that propose to revisit Bayesian statistics from the prediction perspective. The manuscript is however not always completely clear.

I am not a specialist of such matters, but I am puzzled by the presentation of DeFinetti’s result for a finite sequence. It is claimed that \pi is a prior probability whereas I thought that the subtlety is that this is only true for infinite exchangeable sequences. I think this is explained in [1] where it is explained that a general statement requires \pi to be a signed measure. Further it seems to me that the notation is used is rather confusing and not well defined, in contrast with [1]. In the present manuscript \theta is a probability distribution and P_\theta seems to refer to another probability distribution indexed by \theta when I suspect that it is meant that P_\theta is \theta?

Section 3: after reading this several times I think that I have finally understood what the setup and conclusions are. I think communication here could be made more efficient by adopting a less verbose presentation. On p. 7 it is not immediately clear what P and \bar{P} are . It is not clear whether this refers to a generic distribution or a generating process of some sort? In this respect it is not clear whether this covers the regression setup discussed in the introduction and the experimental section. The setup should really be clarified as there are multiple references to a “true distribution” throughout the section.

“The bet reward is anti-symmetric…”: I would not use this term as it may confuse many (including this reviewer) to think that you are referring to other mathematical concepts.

Since the notions of admissible beliefs and minimal bet are so crucial, I would suggest that they have their own displayed definitions to help the reader.

“Restricting to minimal bets hence simply ensures that the agents do not arbitrarily make bets that are not justified on the ground of their beliefs.” My understanding of this is that assuming the agents have access to r (is this stated?) then they do not adapt their strategy to potentially misleading information when in comes to average risk? Either ways this should be made clearer.

In Theorem 5, p seems to be a generic distribution and I do not see any data generating process in the statement. Therefore why put so much emphasis on this in the text? Admittedly the prediction rule is learnt from a distribution but it is unclear what role the properties of this distribution play in the theorem?

There are typos, some of which will unfortunately not be picked-up by a spellchecker…


[1] G. Jay. Kerns & Gábor J. Székely, Definetti’s Theorem for Abstract Finite Exchangeable Sequences

---

### Author Rebuttal · Authors · 2024-05-11

We thank the reviewers for all the insightful feedback, which we believe will significantly enhance the clarity and contribution of our work.

**Clarification of De Finetti’s Theorem:**
We appreciate the opportunity to clarify the presentation of De Finetti's theorem, and explain missing notation. We do indeed call on De Finetti's result for infinite exchangeable sequences, rather than any finite-length variant (although those negative results are certainly interesting!). We incorporated your feedback into this section, and added an appendix section introducing the theorem in more detail, and connecting to the standard presentation. You can see the changes here: https://anonymous.4open.science/r/implicit-bayes-aabi/definetti-rewrite--except.pdf

**Improvements to Section 3:** We agree that the presentation of this section can be improved. The original version of this section can be found in the appendix, but was cut from main paper body for space. We'll aim to incorporate your feedback and make this section more concise while precisely stating the setup. We aim to share a rewritten version of the section before the end of the discussion period.

**Experimental Comparison of a larger array of Bayesian methods:** In response to requests for a broader experimental analysis, we conducted experiments with Bayesian Neural Network (BNN) methods including Laplace approximations, MC Dropout, and HMC sampling. We'll aim to share the new results during the discussion period. In the intermediate term, here are results including the Laplace approximation:
https://anonymous.4open.science/r/implicit-bayes-aabi/regression--big-result-figure--up-to-laplace.pdf

**Performance disparity:** We added an appendix section that digs a bit deeper into where the performance disparity of different methods comes from. You can find it here:
https://anonymous.4open.science/r/implicit-bayes-aabi/what-gives-performance-advantage--exceprt-for-reviewers.pdf

In short, standard MLPs sometimes overfit significantly, and they sometimes do so in a very reproducible way, so that ensembling multiple models doesn’t help.

We also added a version of the results figure that aggregates performance only over training data sizes greater than $70$:
https://anonymous.4open.science/r/implicit-bayes-aabi/regression--big-result-figure--perf_from70_to99.pdf

Again, thank you for all your questions. We'll respond to them in detail during the discussion period, and update the paper accordingly.

---

> ### Author Rebuttal · Authors · 2024-05-11
>
> # Response to Reviewer cHUM
> ### Presentation of De Finetti for finite sequences
>
> Thank you for pointing these details out. Indeed, the presentation lacks some details for completeness, and the notation requires clarification. $P_\theta$ is meant to be an element of $\Theta$, i.e. $P_\theta\in\Theta$, whereas some other works simply use $\theta$ for the probability measure, which wasn’t explained in the paper. We thought seeing $\theta$ as a probability measure might confuse some readers, but we now updated this part and tried to clarify the notation. We have also included a more comprehensive explanation in the appendix. However, to maintain readability in the main text, we have avoided discussing sigma-algebras and related topics wherever possible.
>
> We did intend to call on the ‘standard’ De Finetti result for infinite sequences, not the finite version. We did present the result in a slightly non-standard way by claiming that the sequence $(X_1, X_2, …)$ is exchangeable if and only if the distribution of the finite sequence $(X_1, …, X_n)$ of any length is a mixture (we should have made the “for any n” explicit in the paper). This is indeed different from the presentation in [1, 2], and we agree it warrants some justification. We added this justification now to an appendix. We think presenting it in this way, however, is simpler for a reader not familiar with the details of measure theory, as it pertains to the finite sequence to which one would apply the Bayes rule directly, and it skips over the details of an infinite product space.
>
> We were previously unaware of the negative results for finite exchangeable sequences, and we find these intriguing. We added the reference with appropriate context in a footnote to emphasise that we demand exchangeability of the infinite sequence to claim implicit Bayesianness.
>
> For all the changes mentioned above, you can see the updated version of this part of the paper here (including the more in-depth appendix section on the De Finetti theorem):
> https://anonymous.4open.science/r/implicit-bayes-aabi/definetti-rewrite--except.pdf
>
> Please let us know if you still find any issues with the presentation of the De Finetti’s theorem, and how it relates to statements about distributions of finite sequences!

---

> > ### Author Rebuttal · Authors · 2024-05-11
> >
> > ### Presentation of Section 3 (Adversarial Bets Result):
> > We agree the presentation of this section could be improved. For context: the original version of this section has been moved to Appendix A for lack of space, and we wrote a shorter version that tries to only give an intuition for the results. In retrospect, we acknowledge that we should have maintained clarity in the setup and definitions of terms within the main paper.. We will rewrite this section with your feedback in mind (less verbose, state what is meant by “beliefs”, “data-generating distribution” etc., clarify the setup, and explain how the setup links to the introduction and experimental section,standalone definitions for “admissible”, “minimal”). If time allows, we’ll aim to do so by the end of the rebuttal period and share the new version with the reviewers.
> >
> > > On p. 7 it is not immediately clear what P and \bar{P} are
> >
> > These are just arbitrary probability mass functions, used as placeholders to state the definitions of “favourable” and “admissible”
> > > In Theorem 5, p seems to be a generic distribution and I do not see any data generating process in the statement.
> >
> > $p$ can indeed be any exchangeable distribution. The theorem says that the expected return to an agent (with the expectation taken under $p$) can be negative if the “agent’s beliefs” are not exchangeable. Hence, the interpretation of the theorem is that, no matter what the true data-generating distribution is, as long as it’s exchangeable, the result of the theorem holds and the non-exchangeable agent will lose in expectation.
> >
> > We understand your feedback as there being too much emphasis on the interpretation, and too little on precisely stating the setup. We’ll try to adjust for that.
> >
> >
> > [1] G. Jay. Kerns & Gábor J. Székely, Definetti’s Theorem for Abstract Finite Exchangeable Sequences
> > [2] Edwin Hewitt & Leonard J. Savage, Symmetric Measures on Cartesian Products

---

> > > ### Author Rebuttal · Authors · 2024-05-11
> > >
> > > # Response to reviewer bLoo
> > >
> > > Thank you for the feedback. We appreciate you found the concept interesting, and we will work to incorporate your suggestions to improve its execution.
> > >
> > > > It would have been nice to see more Bayesian Neural network methods in the results to see how Bayesian their implied prediction rules actually are.
> > >
> > > We agree. We conducted additional experiments with the following methods:
> > > - Laplace approximation to BNNs [1]
> > > - MC Dropout
> > > - HMC sampling of the BNN posterior
> > > - Prior Networks [2] with a full MLP (not just a linear model)
> > >
> > > We will share a link to the new results within the rebuttal period once those are complete. An interesting take-away that we observe with these results is that MLPs with the Laplace approximation (with a fixed prior precision) are more implicitly Bayesian than a GP with kernel hyperparameters optimised with marginal likelihood.
> > >
> > > Deep ensemble [3] results are already present in the paper (last paragraph of page 10), which we think highlight an interesting takeaway: they do not seem to help substantially with our measure of implicit Bayesianness. This clearly goes against a popular narrative that deep ensembles are an unreasonably effective Bayesian approximation, which we consider surprising.
> > >
> > > > There don't seem to be any recommendations of how one would use their insights to actually construct methods that are Bayesian. This is fine in itself if it wasn't hinted at in the rest of the paper and in the conclusion
> > >
> > > We will tone down the statements in the introduction/conclusion to try and better reflect the contributions of the paper.
> > >
> > > > The paper claims that the true function was chosen to yield a model mismatch for a GP with a smooth kernel. What is the motivation behind this?
> > >
> > > We wanted to create a setup in which we could observe some interesting trade-offs. If we stuck with smooth, stationary functions, we would likely get the unsurprising result that stationary GPs with an RBF kernel are really good at modelling these. Some of the claimed hypothetical benefits of deep learning over GPs are feature-learning, which can loosely be likened to kernel selection based on observed data, and the fact that they might potentially be able to express “priors” that would be difficult to represent with a GP (e.g. discontinuous functions). If either of these hold, we would hopefully observe that neural networks can yield better predictive performance, while potentially sacrificing implicit Bayesianness.

---

> > > > ### Author Rebuttal · Authors · 2024-05-11
> > > >
> > > > > it is surprising that the GP has better performance than MLPs (Figure 5). Do you know what led to this?
> > > >
> > > > It appears that 1) in the low data regime, or 2) for some particular training data samples, MLPs can overfit significantly and they do so in a reproducible way, so that ensembling multiple models doesn’t seem to help.
> > > > MLPs with the Laplace approximation seem to do much better, at least on par with the GP with fixed kernel hyperparameters, as the Laplace approximation seems to sufficiently counteract the overfitting.
> > > >
> > > > We added an appendix section that dives into this a little bit deeper:
> > > >
> > > > https://anonymous.4open.science/r/implicit-bayes-aabi/what-gives-performance-advantage--exceprt-for-reviewers.pdf
> > > >
> > > > > Section 2.3 is very interesting but it seems that only Definition 1 is used in the rest of the paper, am I correct in thinking that?
> > > >
> > > > Yes, with the possible exception of discussion of future work, where the c.i.d. Condition becomes relevant for Martingale Posteriors. Although only one of the 3 equivalent conditions is used for measuring implicit Bayesianness, we think a contribution of the paper is presenting a somewhat novel perspective for how to think about neural networks as being Bayesian, and we think aggregating the different conditions in one place is useful context to future readers aiming to potentially extend upon this work.
> > > >
> > > > > Methods shown in section 4.2 are known to be approximations to Bayesian methods, so how does the insights from the rest of the paper help here?
> > > >
> > > > To the best our knowledge, weight decay only has a Bayesian approximate inference interpretation when it is scaled as $\propto n^{-1}$ with dataset size $n$. We observe much better measures of implicit Bayesianness with exponents other than $-1$, which do not have a known Bayesian interpretation. We think that’s an interesting result.
> > > >
> > > > > What is the "Joint" in the x and y axis in Figure 2?
> > > >
> > > > The ‘joint’ log-likelihood refers to the log-likelihood for all the labels $y_1, \dots, y_100$ conditioned on the inputs $x_1, \dots, x_100$ obtained by multiplying the probability of $y_n$ conditioned on $y_1, \dots, y_{n-1}, x_1, \dots, x_{n}$ obtained from the prediction rule (i.e. by training on the conditioning set and making a prediction) for all $n=1\dots 100$.
> > > >
> > > > [1] Laplace Redux -- Effortless Bayesian Deep Learning
> > > > [2] Randomized Prior Functions for Deep Reinforcement Learning
> > > > [3] Simple and Scalable Predictive Uncertainty Estimation using Deep Ensembles

---

> > > > > ### Author Rebuttal · Authors · 2024-05-11
> > > > >
> > > > > # Response to reviewer MPvh
> > > > >
> > > > > Thank you for the feedback.
> > > > >
> > > > > We do agree that the proposed measure is simple; this is in large part by design. We do think the motivation behind it and the required change in perspective is more subtle.
> > > > >
> > > > > Providing a concrete strategy for improving deep learning methods is not a goal of this paper, but we certainly think one could build on top of our paper to explore how to improve deep learning methods along our proposed metric of implicit Bayesianness. We think this is interesting future work.
> > > > >
> > > > > To try and address the reviewer’s concern about the level of contribution, we tried to substantiate the contribution of measuring Bayesianness of existing methods in a new way by expanding our experiments to encompass more Bayesian and Bayesian-inspired deep learning methods, including Laplace, MC-Dropout, Prior Networks for MLPs, and Markov-Chain Monte-Carlo sampling (HMC) on an MLP. We will aim to share those before the end of the rebuttal period.

---

> > > > > > ### Author Rebuttal · Authors · 2024-05-11
> > > > > >
> > > > > > # Response to reviewer Y6nz
> > > > > >
> > > > > > Thank you for the feedback! We’ll aim to incorporate it into the paper, and share our new changes wherever possible.
> > > > > >
> > > > > > ### Presentation of Section 3 (Adversarial Bets Result):
> > > > > > We agree the presentation in this section could be improved. For context: the original version of this section has been moved to Appendix A for lack of space, and replaced with a condensed version. In retrospective, we should have still aimed to precisely state the definitions and the setup. We will rewrite this section with your feedback in mind. We’ll aim to do so by the end of the rebuttal period and share the new version with the reviewers.
> > > > > >
> > > > > > ### Complexity of the method:
> > > > > > Thanks for pointing this out. We agree it would be great to state the complexity explicitly. The complexity of our method can be considered to be $O(N P D S)$ where:
> > > > > > - $N$ is the length of the sequence of random variables for which we try to estimate exchangeability
> > > > > > - $P$ is the number of permutation samples considered
> > > > > > - $D$ is the number of samples of sequences from the data-generating distribution
> > > > > > - $S$ is the number of seeds considered
> > > > > > With $N=100, P=10, D=10, S=10$, which is what we use by default, this means $100000$ training runs per method for the evaluation results presented in our paper. We updated the paper to include the method's complexity.

---

> > > > > > > ### Author Rebuttal · Authors · 2024-05-11
> > > > > > >
> > > > > > > Question Answers:
> > > > > > > - “I assume the implicit Bayesianness check will work with even fewer data, such that it becomes even more overparameterized. Is it possible to incorporate some theories from that field to predict the implicit Bayesianness given results on fewer data?” We think that sounds like an exciting direction to explore, although we don’t have any concrete suggestions at the moment.
> > > > > > > - “I would like to know about the relation between the proposed metric vs cross-validation. After reading the work, I view cross-validation as validating the implicit Bayesianness for a single datapoint.”
> > > > > > > I think our measure of performance can be seen as a kind of cross-validation metric, since it sums the log-likelihoods of unseen datapoints over different training sizes, and averages the results over different permutations (different cross-validation folds). The metric of joint log-likelihood is related to (and in the case of exact Bayesian models, the same as) ‘marginal likelihood’ in the context of empirical Bayes, and marginal likelihood has known deep connections to cross-validation [1]. Our measure of implicit Bayesianness in some sense measures the ‘variance‘ of the validation-set performance over different cross-validation folds (different permutations of the data), where the validation set-performance has to be computed in a very specific way, but I’m not sure that’s a helpful way of viewing the metric.
> > > > > > >
> > > > > > > [1] Edwin Fong, Chris Holmes, “On the marginal likelihood and cross-validation”

---

> ### Author Rebuttal · Authors · 2024-05-11
>
> # Response to reviewer XqiK
> Thank you for the feedback and the interesting suggestions and comments!
>
> > Exact Bayesian approaches enjoy various benefits; for instance we can extract epistemic uncertainty estimates (e.g., Kendall & Gal, 2017), or validate a certain model by validating the appropriateness of its prior and likelihood separately.
>
> This is a very interesting question. As for separating epistemic and aleatoric uncertainty, one approach to doing this with prediction rules is to use the Martingale Posteriors [1] method. That procedure is only guaranteed to give a consistent result if the prediction rule is conditionally identically distributed (c.i.d.), and, to our understanding, will only reflect “true” epistemic uncertainty if the method is implicitly Bayesian.
> We do think exploring the connections between theoretical properties of the epistemic uncertainty estimates from Martingale Posteriors and measures of implicit Bayesianness is an exciting avenue for future research.
>
> > is [measuring variance of the log predictive densities] still reasonable if we compare two rules with significantly different values for the expected log loss? [...] the expected log loss incurred by MLPs with different regularization scalings are very different, so it could be an alternative explanation to "non-Bayesian regularization scalings are more 'implicitly Bayesian'".
>
> Whether measuring the variance of the log-predictive densities is the most reasonable approach is a good question. One of the main advantages is that this metric is invariant to differentiable invertible reparameterisations of the data.
> I think another interesting metric could be derived from looking at the adversarial bet robustness result in Theorem 1 and asking: for a fixed size bet (say in L1 norm), what’s the highest expected loss of the prediction rule in this setup?
> I don’t think that the trend of lower expected negative log-joint density leads to lower variance is necessarily to be expected. The negative log-joint density can be negative (and is, in some cases), and so lower values of the negative log-joint density can actually have higher absolute values.

---

> ### Author Rebuttal · Authors · 2024-05-11
>
> > we are not generally interested in the predictive performance for very small values of . Is there any reason not to start the summation from some , or how will the experimental results change if we change the measure in this way?
>
> That’s a good question! We recreated the figure, but with predictive performance metric averaging only over results with a “training data size” of at least 70, and got fairly similar trends. Here is the figure:
>
> https://anonymous.4open.science/r/implicit-bayes-aabi/regression--big-result-figure--perf_from70_to99.pdf
>
> We included the figure in the paper with further explanation. Also, the performance breakdown that we now added to the appendix might be of interest:
>
> https://anonymous.4open.science/r/implicit-bayes-aabi/what-gives-performance-advantage--exceprt-for-reviewers.pdf
>
> We wanted a predictive performance measure that reflects the aggregate performance of a prediction rule over all train sizes, but you’re absolutely right that we’re often interested in performance for larger train sizes.
>
> **Prior Nets on MLPs, and MCMC methods**
>
> We agree that these would be interesting. We are running experiments with HMC and prior networks on MLPs, and will share the results before the end of the rebuttal period. We tried to get the results with SGLD, but it proved impossible to tune it to be stable on our problem.
>
> [1] Edwin Fong, Chris Holmes, Stephen G. Walker, Martingale posterior distributions

---

### Meta-Review · Area_Chair_Fiwe · 2024-05-24

**Recommendation:** Accept (Poster)
**Confidence:** 4

**Metareview:**

The paper considers the benefits of using implicit Bayesian predictions in the context of non-Bayesian models. It characteristics “implicit Bayesianism” as algorithms whose predictions meet certain criteria: exchangeability, stationarity & conditionally independently distributed, and spreadability. It compares several methods by measuring their exchangeability and illustrates the benefits of implicit Bayesianism.

The reviewers agree that the paper is interesting, and that it provides useful insights and experiments to support them. There were some concerns about the presentation at times, and that while the results are compelling, there isn’t a discussion about how to learn from those results to improve current deep learning or Bayesian methods. Presentation concerns seem to have been largely addressed during rebuttal.

Overall, this seems like an insightful paper, which would contribute to an interesting discussion at AABI.

---

### Decision · Program_Chairs · 2024-05-27

Accept